# Group 2 innate lymphoid cells are a non-redundant source of interleukin-5 required for development and function of murine B1 cells

Karoline F. Troch[1,12], Manuel O. Jakob[1,12], Patrycja M. Forster[1], Katja J. Jarick[1], Jonathan Schreiber[2], Alexandra Preusser[1], Gabriela M. Guerra[3], Pawel Durek[3], Caroline Tizian[1], Nele Sterczyk[1], Sofia Helfrich[1], Claudia U. Duerr[1], David Voehringer[4], Mario Witkowski[1,5], David Artis[6,7,8,9,10], Tim Rollenske[2], Andrey A. Kruglov[3], Mir-Farzin Mashreghi[3,11] & Christoph S. N. Klose[1] ✉

Tissue-resident immune cells, such as innate lymphoid cells, mediate protective or detrimental immune responses at barrier surfaces. Upon activation by stromal or epithelial cell-derived alarmins, group 2 innate lymphoid cells (ILC2s) are a rapid source of type 2 cytokines, such as IL-5. However, due to the overlap in effector functions, it remains unresolved whether ILC2s are an essential component of the type 2 response or whether their function can be compensated by other cells, such as T cells. Here we show a non-redundant role of ILC2s in supporting the development and function of B1 cells. We demonstrate that B1 cells fail to develop properly in the absence of ILC2s and identify the IL-33 receptor on ILC2s as an essential cell-intrinsic regulator of IL-5 production. Further, conditional deletion of *Il5* in ILC2s results in defective B1 cell development and immunoglobulin production. Consequently, B1 cells with phosphatidylcholine specific B cell receptor rearrangements are diminished in ILC2-deficient mice. Thus, our data establish an essential function of ILC2s in supporting B1 cells and antibody production at barrier surfaces.

To generate protective antibodies against thymus-dependent antigens, follicular B cells rely on T cell help via the expression of CD40 ligand (CD40L). The interaction between T and B cells constitutes a quality control mechanism of the immune system that ensures the specificity of the immune response by integrating signals from two independent cell types with different activation requirements. Innate lymphoid cells (ILCs) are innate counterparts of T cells. Although ILCs lack the the cell receptor (TCR), these cells have a similar functional diversity in terms of lineage-specifying transcription factors, such as T-bet, GATA-3 and RORγt and with respect to effector functions[1,2]. Group 2 ILCs (ILC2s) depend on GATA-3 and produce type 2 cytokines, such as interleukin (IL)-5, IL-9, IL-13 and Amphiregulin (Areg). While ILC2s were not reported to express CD40L[3], some studies suggest that ILC2s can support thymus-independent antibody production by innate-like B1 cells via the cytokine IL-5[4–6]. However, multiple cell types, including T helper 2 (Th2) cells, were reported to secrete IL-5 and could potentially compensate ILC2-deficiency during a physiologic immune response in lymphoreplete mice, making ILC2 dispensable for B1 cells[7,8]. Indeed, redundant function of ILCs were reported under certain conditions[9,10]. Therefore, whether ILC2 are a non-redundant source of IL-5 for B1 cell development in lymphoreplete mice remains unresolved.

B1 cells have a long lifespan, can self-renew, secrete large amounts of circulating natural IgM antibodies and can be subdivided into CD5+ B1a and CD11b+ B1b cells[11]. B1 antibodies are directed against oxidized lipids and antigens found in apoptotic cells, in the cell wall of gram-positive or gram-negative bacteria, and target phosphatidylcholine, phosphorylcholine or lipopolysaccharide[11].

ILC2s were reported to be the main source of IL-5 in tissues[12] and ILC2-deficient mice have diminished numbers of eosinophils[13], an IL-5-dependent subset of myeloid cells[14,15]. Therefore, we hypothesized that ILC2s are a non-redundant source of IL-5 strictly required to support B1 cells in lymphoreplete mice.

Here we show that ILC2-deficient mice have reduced frequencies and absolute numbers of B1a and B1b cells. Using single-cell sequencing, we further demonstrate that IL-5ra+ B1 cells fail to proliferate, correlating with the downregulation of an array of IL-5-dependent target genes. Furthermore, mice harboring a conditional deletion of *Il5* in ILC2s have reduced B1a and B1b cells, directly linking ILC2-derived IL-5 to the B1 cell phenotype. In contrast, regular proportions of B1 cells developed in mice with conditional deletion of *Il5* in T cells. Similar findings in *Il33*−/− and *Nmur1*iCre-eGFP mice crossed to *Il1rl1*flox/flox mice argue for an essential function of IL-33 as a critical upstream signal for IL-5 production in ILC2s. Finally, we detected reduced serum antibody production and phosphatidylcholine-specific B cell receptor (BCR) rearrangements in the absence of ILC2s. Therefore, our data reveal an innate pathway of antibody production by B1 cells that is strictly dependent on ILC2-derived IL-5 that cannot be compensated by other cell types.

## Results

### B1 cells do not develop properly in the absence of ILC2s
We could recently show that eosinophils do not develop in normal proportions in ILC2-deficient mice[13]. Previous work has demonstrated an essential function of the IL-5 - IL-5ra axis for eosinophil development[14,15] and ILC2s were reported to be the main source of IL-5 in tissues[12]. Since in vitro work suggests that B1 cells may be dependent on IL-5 signals[14,15], we aimed to test whether ILC2s are required for B1 cell development. We have recently introduced a genetic mouse model based on Neuromedin U receptor 1 (*Nmur1*) to genetically target ILC2s, where a P2A-iCre-T2A-eGFP cassette was inserted in exon 2 and 3 of the *Nmur1* gene (referred to as *Nmur1*iCre-eGFP mice)[13,16]. We have generated ILC2-deficient mice by crossing *Nmur1*iCre-eGFP mice to *Id2*flox/flox animals that lack this essential transcriptional repressor for ILC development specifically in ILC2[4,13,17,18]. To test our hypothesis that ILC2 deficiency affects B1 cell development, we analyzed B1 cells in the peritoneal cavity of *Nmur1*iCre-eGFP *Id2*flox/flox mice and Cre-negative littermate controls (*Id2*flox/flox). Indeed, *Nmur1*iCre-eGFP *Id2*flox/flox mice had reduced proportions and absolute cell numbers of B1 cells (gated as IgM+CD23− or CD19+CD23−) in the peritoneal cavity (Figs. 1A, B, S1A). B1 cells can be further subdivided into CD5+ B1a cells and CD5− B1b cells with different developmental trajectories[11]. ILC2-deficiency resulted in a substantial reduction in both, B1a and B1b cells (Fig. 1A, B). Moreover, B1 cells were diminished in *Nmur1*iCre-eGFP *Id2*flox/flox mice in all organs investigated, including the omentum, thoracic cavity, lung and spleen compared to *Id2*flox/flox littermate control mice (Figs. 1C–F, S1A–G). This reduction was independent of the gating strategy used to identify B1 cells, and, thus, is not the consequence of a downregulation of surface markers (Fig. S1A–G). We did not detect a reduction in other immune cell populations investigated in the peritoneal lavage of ILC2-deficient mice in particular not in B2 cells (Fig. S1H–P).

To exclude potential cell-intrinsic effects of our targeting strategy on B1 cells, we performed fate-labelling experiments in *Nmur1*iCre-eGFP*Rosa26*flSTOP-RFP/+ and *Id2*CreErt2/+*Rosa26*flSTOP-YFP/+ mice. While we detected high expression of RFP/YFP reporting *Nmur1* or *Id2* expression in ILC2s, B1 and B2 cells showed no fate-map signal, excluding direct effects of the mouse model on B cells

(Fig. S2A–F). We also did not detect *Nmur1* fate-labeling in eosinophils (Fig. S2D). To additionally dismiss a role of eosinophils in the observed B1 cell reduction, we examined Δ*dblGATA* mice, which lack eosinophils but not ILC2. However, we did not observe reduced proportions of B1 cells in these mice (Fig. S2G–K). Collectively, these data point towards a direct ILC2-dependent function required for B1 cell development or maintenance.

Next, we analyzed B1 cells in relative and absolute B1 cell numbers over time in the peritoneal cavity and observed reduced B1 frequency and counts at all time points analyzed (Fig. 1G). These results suggest that the reduction in B1 cells is not compensated over time. We also confirmed the defective B1a and B1b cell development in *Nmur1*iCre-eGFP *Gata3*flox/flox mice, an alternative strain of ILC2-deficient mice generated in our laboratory[13] (Fig. 1H, I). Finally, we identified clusters of ILC2s and B1 cells in the mesenterium showing a close spatial proximity of these two cell types (Fig. 1J). Coculture of B1 cells with ILC2s or supernatant of stimulated ILC2s promoted B1 cell expansion, suggesting that the close proximity of B1 cells and ILC2 may generate a highly stimulatory milieu (Fig. S2L). Collectively these data suggest that ILC2s and B1 cells sustain a pivotal interaction in tissues.

### IL-5ra+ B1 cells require ILC2 signals for proliferation and survival
To dissect the ILC2-mediated effects on different populations of B cells, we performed single-cell sequencing of peritoneal IgM+ and IgD+ cells from *Nmur1*iCre-eGFP *Id2*flox/flox and littermate control mice (Fig. S3A). In line with the flow cytometry results, markers expressed by B1 cells, such as CD11b and CD5, were underrepresented in *Nmur1*iCre-eGFP *Id2*fl/fl mice compared to controls whereas CD19+ and B220+ cells were equally detected in both groups (Fig. S3B). Unbiased clustering of single-cell transcriptomes revealed 5 clusters with reduced cells in clusters 0-3 in ILC2-deficient mice, whereas clusters 4 and 5 were represented in comparable proportions (Fig. 2A, B). The expression of known B1 cells markers, such as *Anxa2*, *Itgam*, *Ccnd2*, and B2 cell markers, such as *Fcer2* and high expression of *Ptprc*, identified Clusters 0-3 as B1 cells and Clusters 4 and 5 as B2 cells (Fig. 2C)[19,20]. Color-coding of B1 and B2 B-cell clusters showed an evident reduction of B1 cells in *Nmur1*iCre-eGFP *Id2*fl/fl mice compared to control mice, whereas B2 cells were unaffected (Fig. 1, 2D).

Differential-expression analysis of top 30 genes in B1 cells comparing ILC2-deficient and ILC2-sufficient mice revealed downregulation of cell-cycle (*Ccnd2*) and B1 cell-associated genes (*Anxa2*, *Ass1*) in ILC2-deficient mice (Fig. 2E, F). Validation of those, including the top DEG, *Zcwpw1*, in sort-purified B1 cells showed a lower expression in *Nmur1*iCre-eGFP *Id2*fl/fl mice (Fig. 2G). However, B1 cells from *Zcwpw1*−/− mice did not show an obvious phenotype, suggesting *Zcwpw1* is not required for B1 cell development (Fig. S3C). Genes of the cell-cycle, including *Mki67*, were mainly expressed in Cluster 0 and, thus, this cluster represents the proliferative B1 B-cell cluster (Fig. S3D). Herein, chromosome segregation, nuclear division, DNA replication and mitotic cell cycle phase transition were among the top 10 downregulated gene sets in B1 cells of *Nmur1*iCre-eGFP *Id2*fl/fl mice compared to Cre- littermate control mice (Fig. 2H). Likewise, we found down-regulation of *Mki67*, encoding the major proliferative marker Ki-67 (Ki67) (Fig. 2I), suggesting that proliferation is affected in the absence of ILC2s. We could confirm a reduced expression of Ki-67 in B1 cells of *Nmur1*iCre-eGFP *Id2*fl/fl mice compared to controls via intracellular flow cytometry (Figs. 2J, S3E). To delineate the mechanisms of the responsible cytokine-cytokine-receptor signaling cascade, we identified the expression of cytokine-receptors in B1 and B2 cells. B1 cells expressed several receptors of the innate immune system, such as *Il5rα*, *Il6st* of *Il4rα* (Figs. 2K, S3F). Furthermore, the mean expression of *Il5rα* was enriched in cluster 0 and 2, both of which were strongly diminished in ILC2-deficient mice (Fig. S3G). Due to the importance of IL-5rα signals and the particular reduction of IL5rα+ B1 cells in ILC2-deficient mice (Fig. 2L), we analyzed published IL-5-dependent gene

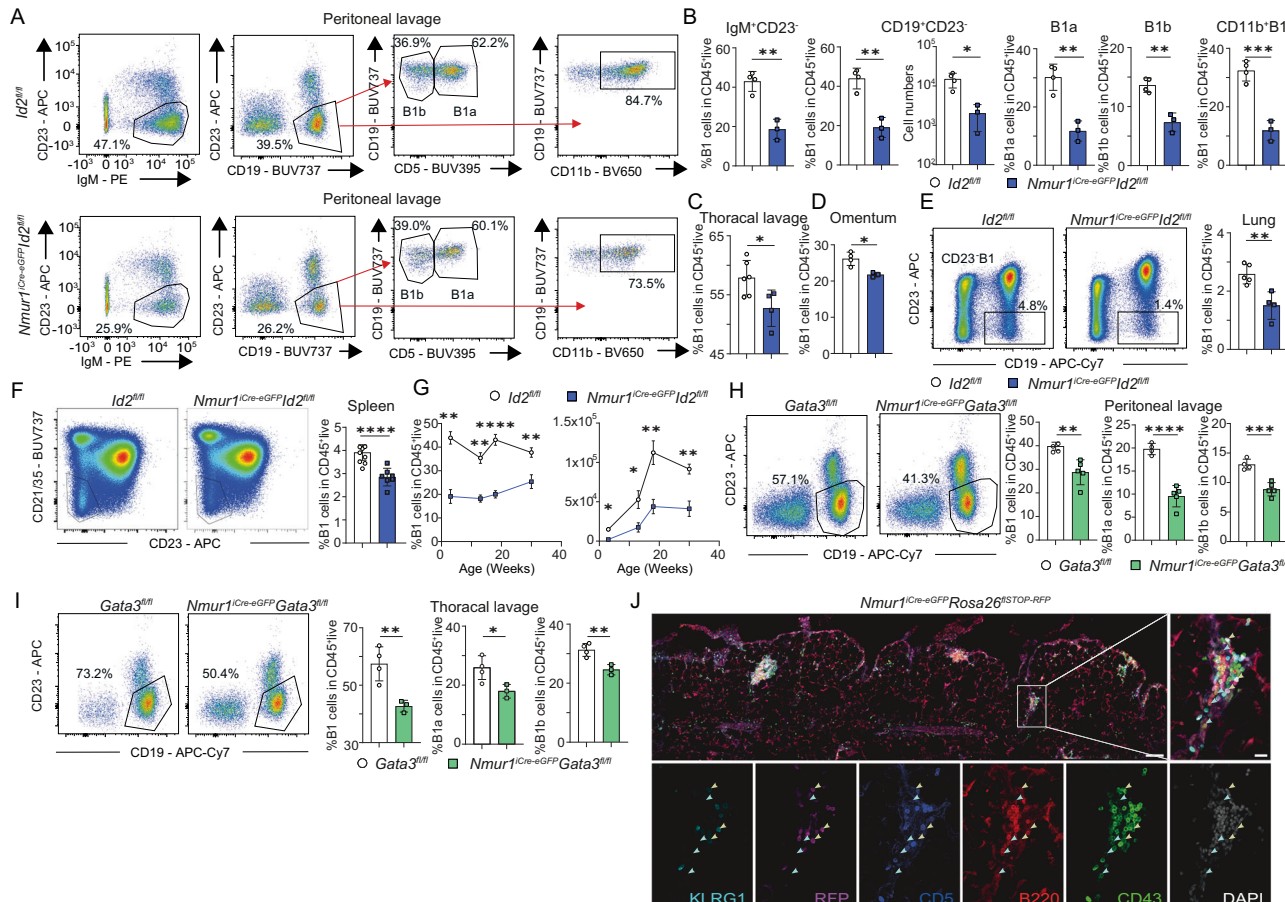

**Fig. 1 | ILC2s shape the B1 cell pool. a** Flow cytometric plots of B1 cells in *Nmur1*[iCre-eGFP] *Id2*[fl/fl] (blue) and littermate *Id2*[fl/fl] mice (white) in the peritoneal cavity in steady state. B1 cells were pre-gated on live CD45⁺ TCRβ⁻. First gate shows B1 cells gated as IgM⁺ CD23⁻. Second gate shows B1 cells gated as CD19⁺ CD23⁻, further gating of B1 cells included CD11b. B1a and B1b cells were subdivided using the marker CD5. **b** Quantification of B1 cells as in (**a**) (*Id2*[fl/fl] *n* = 4, *Nmur1*[iCre-eGFP] *Id2*[fl/fl] *n* = 3). **c–f** Flow cytometric plots and quantification of B1 cells in the (**c**), Thoracal lavage (*Id2*[fl/fl] *n* = 6, *Nmur1*[iCre-eGFP] *Id2*[fl/fl] *n* = 4), **d** Omentum (*Id2*[fl/fl] *n* = 4, *Nmur1*[iCre-eGFP] *Id2*[fl/fl] *n* = 3) **e** Lung (*Id2*[fl/fl] *n* = 5, *Nmur1*[iCre-eGFP] *Id2*[fl/fl] *n* = 4) and (**f**) spleen (*Id2*[fl/fl] *n* = 8, *Nmur1*[iCre-eGFP] *Id2*[fl/fl] *n* = 7). **g** Relative and absolute cell numbers of B1 cells across different ages in *Nmur1*[iCre-eGFP] *Id2*[fl/fl] (blue, 3 weeks *n* = 3, 13 weeks *n* = 4, 18 weeks *n* = 5, 30 weeks *n* = 4) and littermate *Id2*[fl/fl] mice (black, 3 weeks *n* = 4, 13 weeks *n* = 4, 18 weeks *n* = 4, 30 weeks *n* = 5). **h, i** Flow cytometric plots of B1 cells in *Nmur1*[iCre-eGFP]

*Gata3*[fl/fl] (green) and littermate *Gata3*[fl/fl] (white) mice in the (**h**), peritoneal cavity (*Gata3*[fl/fl] *n* = 4, *Nmur1*[iCre-eGFP] *Id2*[fl/fl] *n* = 5) and (**i**), thoracal lavage (*Gata3*[fl/fl] *n* = 4, *Nmur1*[iCre-eGFP] *Id2*[fl/fl] *n* = 3) at steady state. **j** Histology section of omental Fat-Associated-Lymphoid-Clusters (FALCs) from the ILC2 reporter mouse *Nmur1*[iCre-eGFP] *Rosa26*[flSTOP-RFP/+] (violet signal) stained with anti-KLRG1 (light-blue), anti-CD5 (dark blue), anti-B220 (red), anti-CD43 (green), and DAPI. Yellow arrows identify ILC2s (RFP⁺ KLRG1⁺ CD5⁻), blue arrows point towards B1 cells (B220[low], CD43⁺). Scale bar overview 100 μm, zoom 20 μm. Each symbol represents data from one mouse, mean +/− SD, all data are representative of at least two independent experiments. Statistical significance was determined by two-tailed unpaired Student's *t*-test (**b–i**), *$p < 0.05$ **$p < 0.01$, ***$p < 0.001$, ****$p < 0.0001$. Source data, including exact *p*-values, are provided as a Source data file Fig. 1.

sets in our single-cell sequencing data[21]. Consistent with IL-5 as a crucial downstream signal, most known IL-5 target genes were downregulated in B1 cells of *Nmur1*[iCre-eGFP] *Id2*[fl/fl] mice and were not expressed in B2 cells (Fig. 2M). Furthermore, we could confirm a reduced expression of the genes *Ass1*, *Anxa2*, *Ccnd1* and *Zcwpw1* via *qPCR in* sort-purified B1 cells from *Il5*[cre/cre] mice suggesting a common downstream signaling in ILC2-deficient and *Il5*[cre/cre] mice (Fig. 2N). In total, these data suggest that B1 cells receive a decisive signal for survival and proliferation from ILC2s, most likely via the IL-5rα.

**Conditional deletion of *Il5* in ILC2s results in a defective B1 cell development**

The reduction in IL-5ra⁺ B1 cells and IL-5 target genes in B1 cells in ILC2-deficient mice suggests ILC2s as an essential source of IL-5 (Figs. 1, 2L, M). To test whether IL-5 drives the B1 cell phenotype, we performed flow-cytometry of B1 cells in *Il5*[cre/cre] mice. Indeed, B1 cells were reduced in relative and absolute cell numbers in absence of IL-5 confirming previous reports[14,15] (Figs. 3A, S4A, B). Further, we

observed a close spatial proximity of IL-5⁺ ILC2s with B1 cells in omental FALCs, suggesting direct interaction (Fig. 3B). To directly test whether ILC2 are a non-redundant source of IL-5, we generated *Nmur1*[iCre-eGFP] *Il5*[fl/fl] mice to conditionally delete *Il5* in ILC2s. To validate successful deletion of *Il5* in ILC2s but not in T cells, we sort-purified ILC2s and T cells from *Nmur1*[iCre-eGFP] *Il5*[fl/fl] mice and littermate controls and performed qPCR for type 2 cytokines. Indeed, *Il5* but not *Il13* or *Il4* mRNA was diminished in ILC2s in *Nmur1*[iCre-eGFP] *Il5*[fl/fl] mice compared to control mice (Fig. 3C, S4C). CD4⁺ T cells did not show reduced expression of *Il5* upon conditional deletion of *Il5* using *Nmur1*[iCre-eGFP], excluding a major contribution of T cells to the phenotype (Fig. S4D). PMA and ionomycin stimulation of ILC2s in vitro resulted in IL-5 production in ILC2s of control mice but not of *Nmur1*[iCre-eGFP] *Il5*[fl/fl] mice whereas ILC2s of both genotypes produced IL-13 protein confirming specific targeting (Fig. 3D). Furthermore, ILC2s per se were not affected by the deletion of *Il5* in the small intestine or MLN (Fig. 3E, F). Therefore, *Nmur1*[iCre-eGFP] *Il5*[fl/fl] mice are a suitable model to investigate the effects of ILC2-derived IL-5 in vivo.

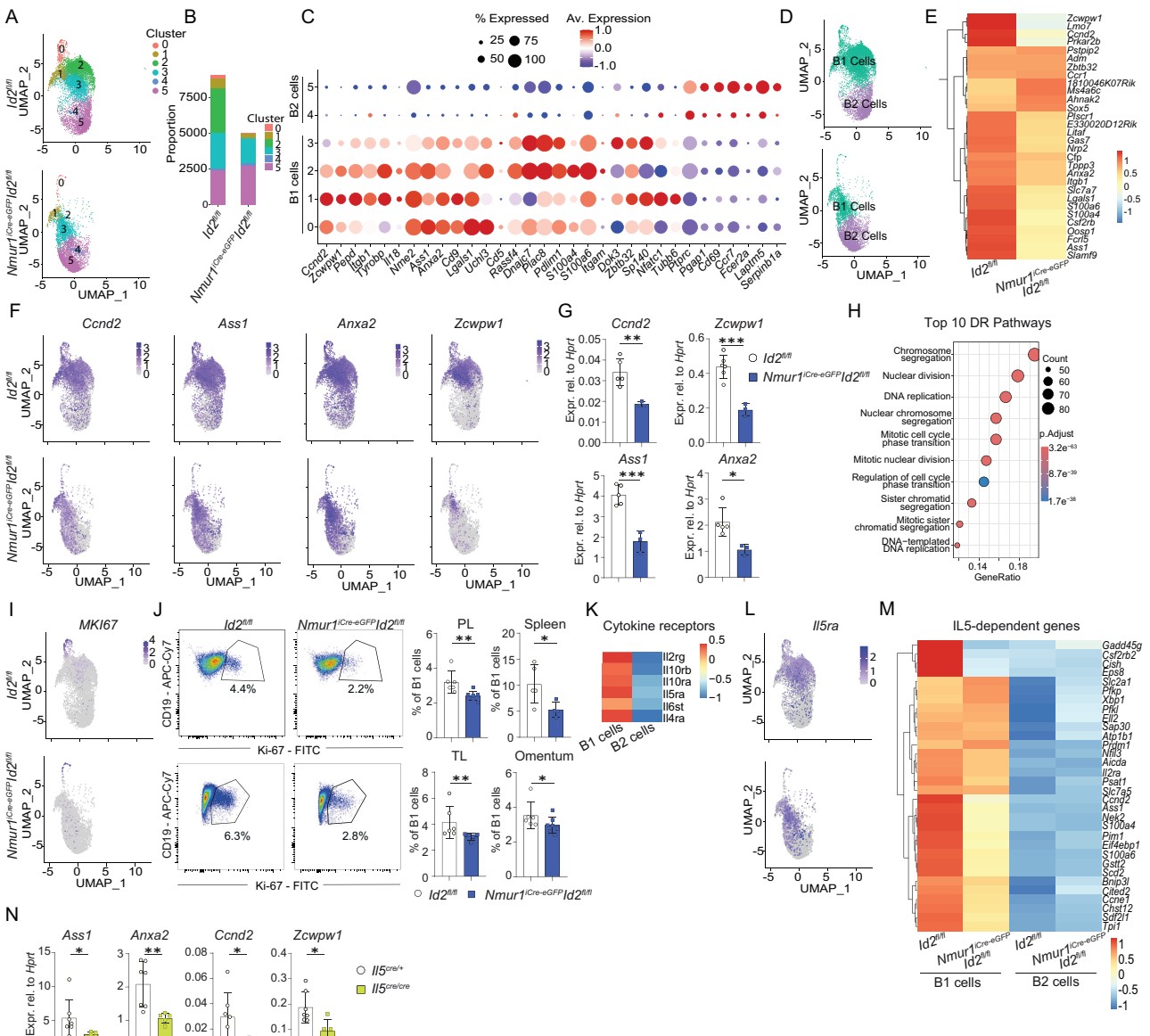

**Fig. 2 | ILC2s expand IL5ra⁺ B1 cells. a** Clustering of flow-sorted and sequenced B cells at a resolution 0.3. **b** Proportions of the clusters of *Nmur1*[iCre-eGFP] *Id2*[fl/fl] vs. littermate *Id2*[fl/fl] mice. **c** B1- and B2-specific genes in each Cluster were plotted according to their average expression and percentage of expression. **d** Annotated and color-coded B1 vs. B2 cells. **e** Heatmap of top 30 differentially-expressed genes in B1 cells of *Nmur1*[iCre-eGFP] *Id2*[fl/fl] vs. littermate *Id2*[fl/fl] mice. **f** UMAPs of *Ccnd2*, *Ass1*, *Anxa2*, *Zcwpw1*. **g** Validation by qPCR of downregulated B1 cell genes. B1 cells were sort-purified from the peritoneal cavity (*Id2*[fl/fl] *n* = 5, *Nmur1*[iCre-eGFP] *Id2*[fl/fl] *n* = 3). **h** Top 10 differentially-regulated pathways in Cluster 0 of the UMAP in a. **i** UMAP showing *Mki67* in B cells, **j** flow cytometric plots and quantification of Ki67⁺ B1 cells (Pre-gated on Live CD45⁺ CD19⁺ CD23⁻) in *Nmur1*[iCre-eGFP] *Id2*[fl/fl] (blue, PL, peritoneal lavage *n* = 7, TL, thoracal lavage *n* = 5, Spleen *n* = 4, Omentum *n* = 7) and littermate *Id2*[flox/flox]

mice (white, PL *n* = 6, TL *n* = 6, Spleen *n* = 5, Omentum *n* = 6). **k** Expression of cytokine-receptors on B1 and B2 cells in ILC2-sufficient mice. **l** UMAP of *IL5ra*⁺ B cells. **m** differentially- regulated gene expression of reported IL-5-induced B1 cell genes[21] of *Nmur1*[iCre-eGFP] *Id2*[fl/fl] vs. (blue) and littermate *Id2*[flox/flox] mice. **n** B1 cells were sort-purified from *Il5*[Cre/Cre] (yellow, *n* = 5) and littermate *Il5*[Cre/+] mice (white, *n* = 7) and qPCR was performed for the indicated genes. Each symbol represents data from one mouse, mean +/− SD, data in (**g**, **j**, **n**) are representative of two independent experiments. Statistical significance was determined by two-tailed unpaired Student's *t*-test (**g**) or Mann Whitney test (**j**, **n**), *\*p* < 0.05 \*\**p* < 0.01, \*\*\**p* < 0.001, \*\*\*\**p* < 0.0001. Source data, including exact *p*-values, are provided as a Source data file Fig. 2.

Thus, we investigated B1 cells in *Nmur1*[iCre-eGFP] *Il5*[fl/fl] mice using comparable readouts by flow cytometry as done in previous experiments. *Nmur1*[iCre-eGFP] *Il5*[fl/fl] mice showed reduced relative and absolute B1 cell numbers in all organs investigated and independent of the B1 cell markers used (Fig. 3G–L). In line with the data obtained in ILC2-deficient mice, B1 cells were already reduced in two-week-old *Nmur1*[iCre-eGFP] *Il5*[fl/fl] mice, suggesting that ILC2s need to deliver IL-5 early during development (Fig. 3M). These data are in accordance with results obtained in ILC2-deficient and *Il5*[cre/cre] mice and demonstrate a causal requirement for ILC2-derived IL-5 (Figs. 1, 3A). As a

complementary approach, we crossed *Cd4*[Cre] *Il5*[fl/fl] mice, to genetically ablate IL-5 in T cells, which were described as producers of IL-5 as well. However, *Cd4*[Cre] *Il5*[fl/fl] mice did not show significant differences in B1 cell proportions compared to littermates control mice (Fig. 3N, O). Therefore, we concluded that ILC2s instead of T cells are the non-redundant source of IL-5 required for the development of B1 cells.

Since B1 cells also expressed receptors for other ILC2 cytokines, such as for IL-4 and IL-6 (Fig. 2M), we tested whether conditional deletion of *Il4/13*, *Il6* and *Areg* in ILC2s affects proper B1 cells

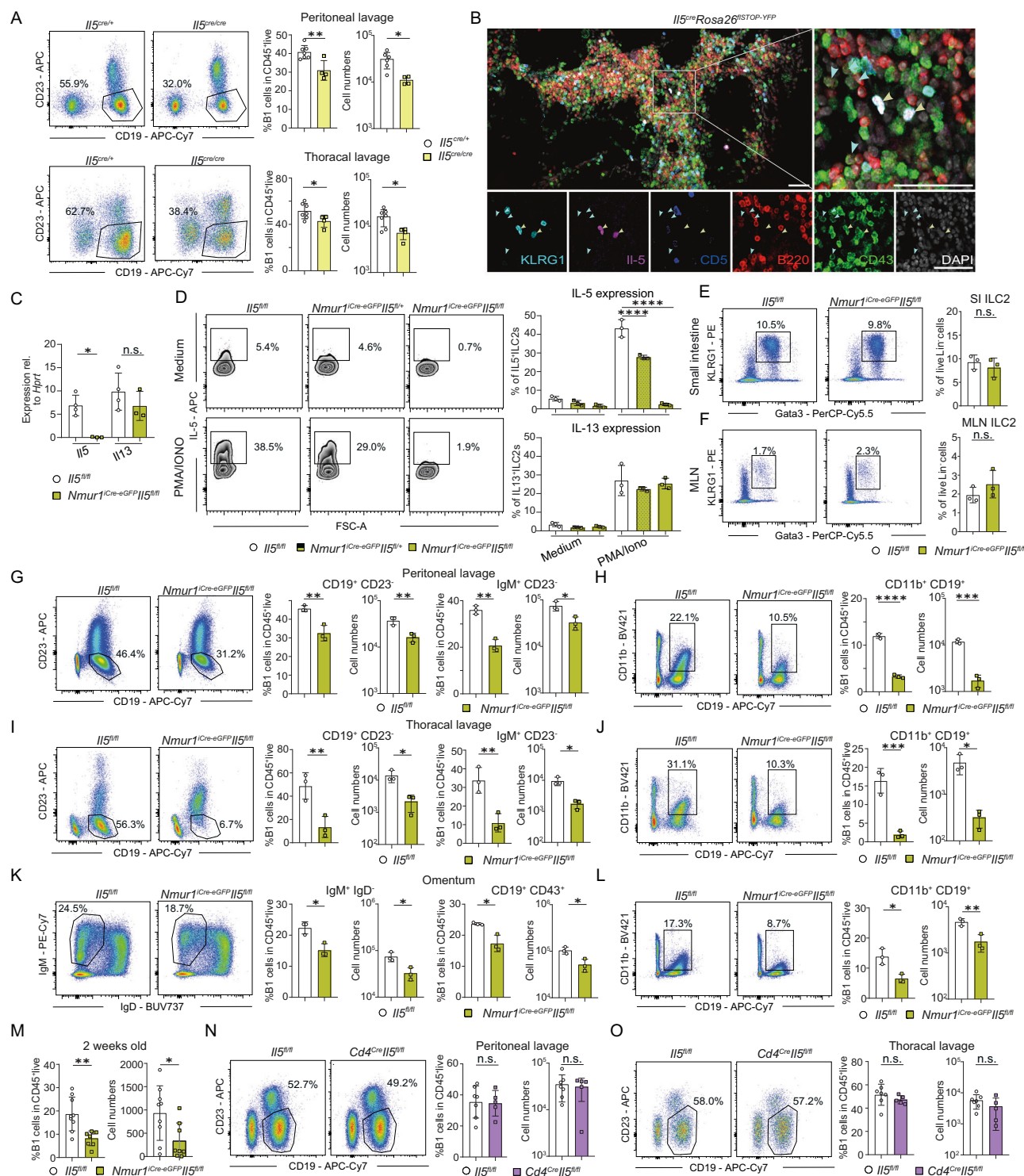

expansion. However, we did not observe a B1 cell reduction in any of the other conditional cytokine knockouts, arguing for a specific effect of ILC2-derived IL-5 (Fig. S4E–G). Taken together, our data from conditional deletion of IL-5 in ILC2s support the role of ILC2s as an essential source of IL-5 for B1 cell development, a function that cannot be compensated by other cell types.

## B1 cells with phosphatidylcholine-specific BCR rearrangements are reduced in the absence of ILC2s

A substantial fraction of B1 cell antibodies is directed against phosphatidylcholine (PtC), which can protect from bacterial infections.

Anti-PtC antibodies are mainly encoded by the $V_H11$ and $V_H12$ immunoglobulin heavy chain variable region gene families[22]. Therefore, we aimed to explore how the BCR repertoire changes in the absence of ILC2s with a particular focus on anti-PtC rearrangements. To this end, we performed single-cell RNA sequencing combined with B cell receptor sequencing of B cells from the peritoneal cavity. Single-cell sequencing recapitulated a similar clustering and gene enrichment as in our previous experiment (Fig. 4A–C, S5A). In particular, B1 cells and IL-5rα+ B cells were reduced (Fig. 4C, D) and the expression of IL-5-induced genes was diminished in *Nmur1*[iCre-eGFP] *Id2*[fl/fl] mice (Fig. S5B). As expected, the expression of *Ighm* (IgM) was detected in all B cells,

**Fig. 3 | ILC2-derived IL-5 is required for proper B1 cell development. a** Flow cytometric plots and quantification of B1 cells in *Il5*^Cre/Cre^ (yellow, *n* = 4) and littermate *Il5*^Cre/+^ mice (white, *n* = 7) in the peritoneal cavity and thoracal lavage at steady state. **b** Histology section of omental FALCs from *Il5*^Cre/+^ *Rosa26*^flSTOP-YFP/+^ mice. Staining with anti-B220 (red), anti-CD43 (green), anti-KLRG1 (light-blue), anti-CD5 (blue), endogenous IL-5 (Violet) and DAPI. Il-5 producing ILC2s (yellow arrows, Il5+KLRG1+CD5-) reside in close proximity to B1 cells (blue arrows, B220^low^CD43+). Scale bars 50 μm. **c** Validation of the mouse line *Nmur1*^iCre-eGFP^ *Il5*^fl/fl^ by qPCR of sort-purified ILC2s from the lung (*Il5*^fl/fl^ *n* = 4, *Nmur1*^iCre-eGFP^ *Il5*^fl/fl^ *n* = 3). **d** IL-5 and Il-13 secretion was determined by flow cytometry of cultured immune cells isolated from the small intestine of *Nmur1*^iCre-eGFP^ *Il5*^fl/fl^ mice and littermate controls, stimulated with PMA/ionomycin (PMA/Iono) for 4 h (*Il5*^fl/fl^ *n* = 3, *Nmur1*^iCre-eGFP^ *Il5*^fl/fl^ *n* = 3, *Nmur1*^iCre-eGFP^ *Il5*^fl/fl^ *n* = 3). **e, f** Relative ILC2 numbers in the (**e**), small intestine and (**f**),

mesenteric lymph nodes (MLN) (*Il5*^fl/fl^ *n* = 3, *Nmur1*^iCre-eGFP^ *Il5*^fl/fl^ *n* = 3). **g–l** Flow cytometric plots and quantification (*Il5*^fl/fl^ *n* = 3, *Nmur1*^iCre-eGFP^ *Il5*^fl/fl^ *n* = 3) of (**g, i, k**), B1 cells (cells were pre-gated on live CD45+ TCRβ-) and (**h, j, l**), CD11b+ B1 cells in the (**g, h**) peritoneal lavage, (**i, j**), the thoracic lavage and (**k, l**), the omentum. **m** Flow cytometric quantification of B1 cells in 2 weeks old *Nmur1*^iCre-eGFP^ *Il5*^fl/fl^ mice and littermate controls (*Il5*^fl/fl^ *n* = 9, *Nmur1*^iCre-eGFP^ *Il5*^fl/fl^ *n* = 8). **n, o** Flow cytometric plots and quantification of B1 cells in the (**n**), peritoneal cavity and the (**o**), thoracal lavage in littermate, co-housed *Cd4*^Cre^ *Il5*^fl/fl^ mice and controls (*Il5*^fl/fl^ *n* = 7, *Cd4*^Cre^ *Il5*^fl/fl^ *n* = 5). Each symbol represents data from one mouse, mean +/− SD, all data are representative of at least two independent experiments. Statistical significance was determined by two-tailed unpaired Student's *t*-test (**a, e–o**) or one-way ANOVA (**c, d**), n.s. non-significant, *$p < 0.05$, **$p < 0.01$, ***$p < 0.001$, ****$p < 0.0001$. Source data, including exact *p*-values, are provided as a Source data file Fig. 3.

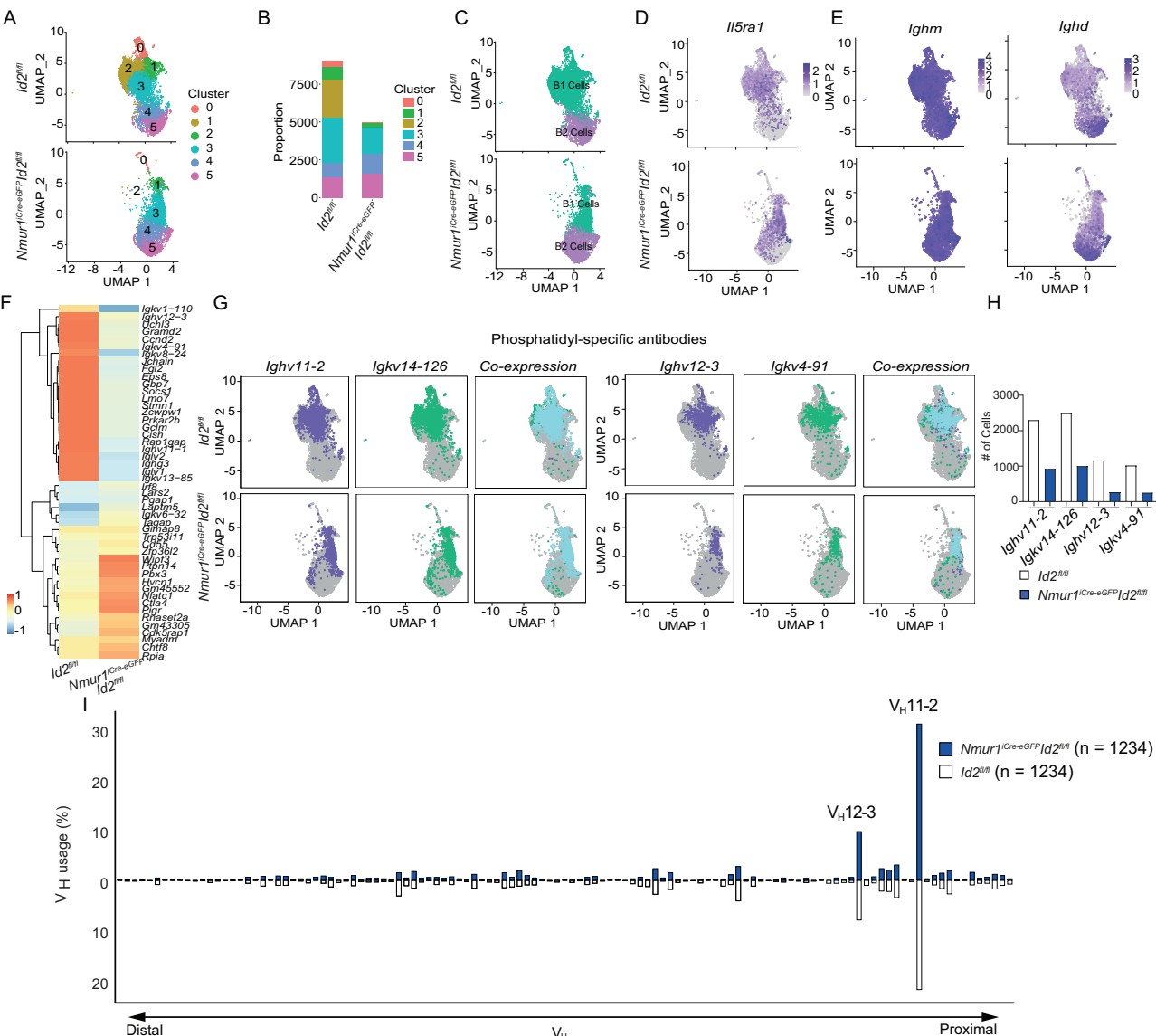

**Fig. 4 | ILC2s support the expansion of phosphatidylcholine specific B1 cells. a** Clustering of flow-sorted and sequenced B cells from the peritoneal lavage at a resolution 0.3. **b** Proportions of the clusters of *Nmur1*^iCre-eGFP^ *Id2*^fl/fl^ vs. littermate *Id2*^fl/fl^ mice. **c** Annotated and color-coded B1 and B2 cells in the peritoneal lavage. **d** UMAP of IL5ra+ B cells. **e** UMAPs of *Ighm* and *Ighd*. **f** Heatmap of differentially-regulated genes including BCR genes comparing *Nmur1*^iCre-eGFP^ *Id2*^fl/fl^ and littermate

*Id2*^fl/fl^ mice. **g** UMAPs and (**h**), quantification of the most important phosphatidyl-specific heavy- and corresponding light chains, *Ighv11-2, Igkv14-126* and *Ighv12–3, Igkv4-91* in *Nmur1*^iCre-eGFP^ *Id2*^fl/fl^ and littermate *Id2*^fl/fl^ mice. **i** *Ighv* usage in single B1 cells from 5 mice from *Nmur1*^iCre-eGFP^ *Id2*^fl/fl^ *(blue)* and littermate *Id2*^fl/fl^ mice (white). *Ighv* gene segments are ordered by their relative proximity to the D segments. Number of cells is depicted in the Figure.

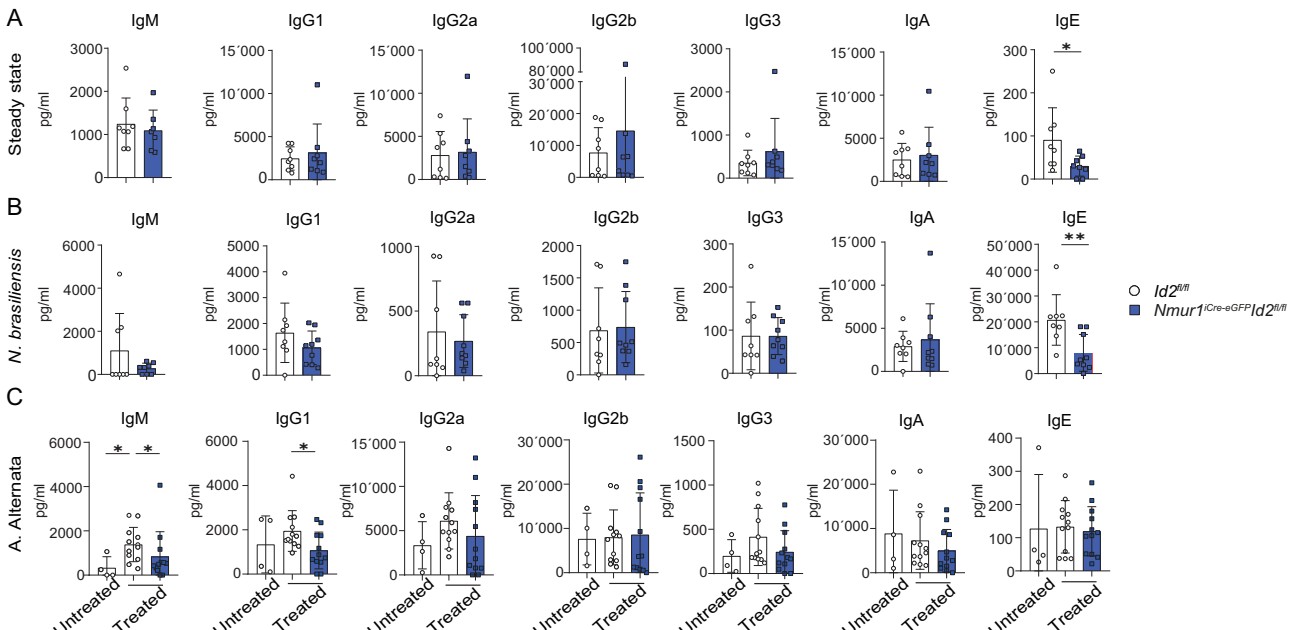

**Fig. 5 | Serum IgM and IgE levels are dependent on the presence of ILC2s.**
**a**−**c** Immunglobulin concentrations in the serum of *Nmur1*[iCre-eGFP] *Id2*[fl/fl] and litter-mate *Id2*[fl/fl] mice were determined by using the LEGENDplex multiplex beads-based assay. **a** Total immunoglobulin concentration in steady state (*Id2*[fl/fl] *n* = 8, *Nmur1*[iCre-eGFP] *Id2*[fl/fl] *n* = 8). **b** Mice were infected with *N. brasiliensis* and serum was harvested at d11 post-infection (*Id2*[fl/fl] *n* = 8, *Nmur1*[iCre-eGFP] *Id2*[fl/fl] *n* = 9). **c** Inflammation was induced by intranasal administration of *Alternaria alternata* (*A. alternata*) extract. Sera were harvested at d7 after induction of inflammation (Untreated *Id2*[fl/fl] *n* = 4, *A. alternata* *Id2*[fl/fl] *n* = 12, *A. alternata Nmur1*[iCre-eGFP] *Id2*[fl/fl] *n* = 12). Each symbol represents data from one mouse, mean +/− SD, data are pooled from two (**a**, **b**) or three (**c**) independent experiments. Statistical significance was determined by two-tailed unpaired Mann Whitney test (**a**–**c**), \**p* < 0.05, \*\**p* < 0.01. Source data, including exact *p*-values, are provided as a Source data file Fig. 5.

whereas *Ighd* (IgD) was restricted to B2 cells (Fig. 4E). In addition, we confirmed the B1 cell clusters by their clear enrichment for $V_H11$ and $V_H12$ usage (Fig. S5C). DEG-analysis revealed down-regulation of specific *Ighv* and *Igkv* genes in ILC2-deficient mice (Fig. 4F). In particular, cells expressing the *Ighv11-2 / Igkv14-126* and *Ighv12-3 / Igkv 4-91* BCR rearrangements were underrepresented in B1 cells of *Nmur1*[iCre-eGFP] *Id2*[fl/fl] compared to control mice and mainly enriched in the IL-5rα⁺ cluster 2 compared to the IL-5rα⁻ cluster 3 (Figs. 4G, H, S5D, E). Next, we aimed to identify, if this reduction of PtC Ig gene transcripts was due to a reduction of PtC-reactive antibodies within the BCR repertoire or due to the reduced number of B1 cells in ILC2-deficient mice. Thus, we aligned the number of cells of both genotypes and compared their *Ighv* usage. We did not detect major differences or shifts within the antibody repertoire (Fig. 4I), comparing *Nmur1*[iCre-eGFP] *Id2*[fl/fl] and control mice. Therefore, our data strongly argue for a non-redundant role for ILC2s in development and maintenance of the B1 cell compartment and the total amount of anti-PtC antibodies, but the B1 BCR specificity seems unaffected by the absence of ILC2s.

## ILC2-deficient mice produce less total IgM and IgE during type 2 immune models

To assess whether the lack of ILC2s affects the extent of antibody production, we determined all isotypes in the serum of *Nmur1*[iCre-eGFP] *Id2*[fl/fl] and control mice at steady-state. While most isotypes were not reduced, we detected a significant reduction of IgE in *Nmur1*[iCre-eGFP] *Id2*[fl/fl] compared to littermate control mice (Fig. 5A). To test whether ILC2s might promote the level of IgM or class switch recombination, we challenged the mice with *Nippostronglyus brasiliensis (N. brasiliensis)* and *Alternaria Alternata* (*A. Alternata*), two well-established models for type 2 inflammation. The infection with *N. brasiliensis* promoted the induction of IgE and ILC2-deficient mice showed reduced levels of IgE, whereas other isotypes remained unchanged (Fig. 5B). This is in line with reports showing that during infection with

*N. brasiliensis*, B1 cells are the main source of non-protective IgE[23]. In the *A. Alternata* model, we observed an induction of total IgM in the serum in ILC2-sufficient mice while a lack of ILC2s reduced the IgM titer (Fig. 5C). Therefore, our data argue that ILC2s are necessary to support the production of IgM and IgE.

## IL-33 - IL-33 receptor mediated activation of ILC2s is required to support B1 cell development

Several publications have shown that the alarmins IL-33, IL-25 and TSLP are dispensable for ILC2 development but are required for cytokine production of ILC2s. Hence, reduced type 2 cytokine production was reported for different strains and combinations of alarmin-deficient mice[24–28]. To identify the upstream regulators required for IL-5 production in ILC2s supporting B1 cells, we analyzed B1 cells in IL-33-, IL-25R- (*Il17rb*⁻/⁻) and TSLPR-deficient mice. We found a reduction of B1 cells in *Il33*⁻/⁻ mice but not in *Il17rb*⁻/⁻ or *Tslpr*⁻/⁻ mice suggesting that IL-33 is the dominant alarmin controlling IL-5 in ILC2s at steady state in different organs (Figs. 6A–D, S6A–F). Furthermore, we detected a reduced expression of Ki-67 via flow cytometry in *Il33*⁻/⁻ mice similar to the data found in ILC2-deficient mice suggesting that IL-33 controls the size of the B1 cell pool (Fig. 6E, F). However, since the IL-33 receptor is expressed by many cells and direct effects of IL-33 on B1 cells have been described before[29], we crossed *Nmur1*[iCre-eGFP] mice to *Il1rl1*[fl/fl] to genetically ablate the IL-33 receptor subunit ST2 in ILC2s[25]. In line with a cell-intrinsic requirement of ST2 in ILC2, B1 cells were diminished in *Nmur1*[iCre-eGFP] *Il1rl1*[fl/fl] mice (Fig. 6G–J). To investigate the downstream effects of alarmin signaling in ILC2s, we sort-purified ILC2s from several knockout lines and performed qPCR. Consistent with a role of IL-33 in triggering IL-5 production in ILC2s via ST2, ILC2 from *Il33*⁻/⁻ and *Nmur1*[iCre-eGFP] *Il1rl1*[fl/fl] mice but not from *Il17rb*⁻/⁻ or *Tslpr*⁻/⁻ mice had reduced *Il5* expression (Fig. 6K). Therefore, we concluded that ILC2-sense IL-33 directly to produce IL-5 and to maintain B1 cells development.

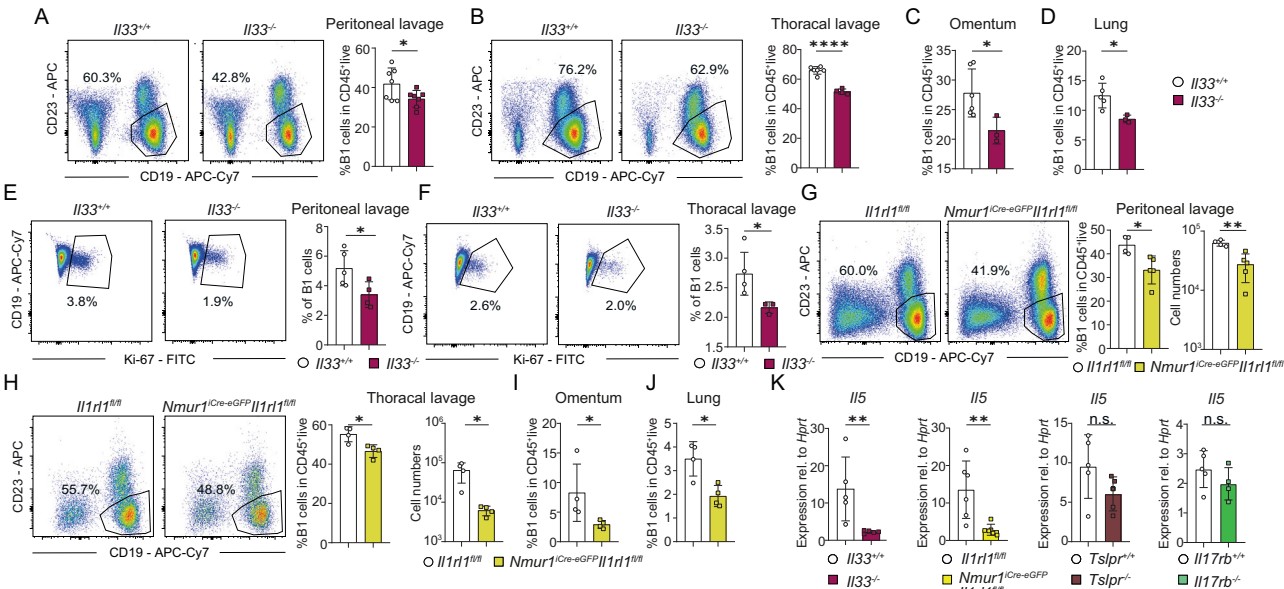

**Fig. 6 | IL-33 triggers IL-5 production in ILC2 to stimulate B1 cells. a–d** Flow cytometric plots and quantification of B1 cells in (**a**), the peritoneal lavage (*Il33*^+/+^ *n* = 7, *Il33*^-/-^ *n* = 7), **b** the thoracal lavage (*Il33*^+/+^ *n* = 6, *Il33*^-/-^ *n* = 4), **c** omentum (*Il33*^+/+^ *n* = 6, *Il33*^-/-^ *n* = 3) and (**d**) lung (*Il33*^+/+^ *n* = 5, *Il33*^-/-^ *n* = 3) of *Il33*^-/-^ and control mice. **e, f** Flow-cytometric analysis of Ki-67 of B1 cells (Pre-gated on Live CD45^+^ CD19^+^ CD23^-^) in *Il33*^-/-^ versus control mice in the (**e**), peritoneal lavage (*Il33*^+/+^ *n* = 5, *Il33*^-/-^ *n* = 4) and the (**f**) thoracal lavage (*Il33*^+/+^ *n* = 4, *Il33*^-/-^ *n* = 3). **g–j** Flow-cytometric plots and quantification of B1 cells of *Nmur1*^iCre-eGFP^ *Il1rl1*^fl/fl^ (ST2) mice and littermate controls in (**g**), the peritoneal lavage (*Il1rl1*^fl/fl^ *n* = 4, *Nmur1*^iCre-eGFP^ *Il1rl1*^fl/fl^ *n* = 5) **h** thoracal lavage (*Il1rl1*^fl/fl^ *n* = 4, *Nmur1*^iCre-eGFP^ *Il1rl1*^fl/fl^ *n* = 4), **i** omentum (*Il1rl1*^fl/fl^ *n* = 4,

*Nmur1*^iCre-eGFP^ *Il1rl1*^fl/fl^ *n* = 4), and (**j**) lung (*Il1rl1*^fl/fl^ *n* = 4, *Nmur1*^iCre-eGFP^ *Il1rl1*^fl/fl^ *n* = 4). **k** Sort-purified small intestinal ILC2s gated as Live CD45^+^ Lin^-^ (CD3, CD5, CD19, Fcεrl, Ly6G) CD127^+^ KLRG1^+^ of *Tslpr*^-/-^ (*Tslpr*^+/+^ *n* = 5, *Tslpr*^-/-^ *n* = 5), *Il17rb*^-/-^ (*Il17rb*^+/+^ *n* = 5, *Il17rb*^-/-^ *n* = 4)⁻, *Il33*^-/-^ (*Il33*^+/+^ *n* = 5, *Il33*^-/-^ *n* = 5) and *Nmur1*^iCre-eGFP^ *Il1rl1*^fl/fl^ (*Il1rl1*^fl/fl^ *n* = 6, *Nmur1*^iCre-eGFP^ *Il1rl1*^fl/fl^ *n* = 7) and qPCR for *Il5*. Each symbol represents data from one mouse, mean +/− SD, data are representative of at least two independent experiments. Statistical significance was determined by two-tailed unpaired Student's *t*-test (**a–j**) or Mann Whitney test (**k**), n.s. non-significant, *$p < 0.05$, **$p < 0.01$, ***$p < 0.001$. Source data, including exact *p*-values, are provided as a Source data file Fig. 6.

## Discussion

To which degree the overlap in effector molecules of ILC2s and other immune cells determines complementary and redundant functions of these cell types is a fundamental research question in the field today. Limitations in genetic targeting made it difficult to experimentally address the question of ILC redundancy. Although ILCs and T cells secrete a similar array of cytokines, the regulation and kinetics of T cell and ILCs response may vary. Recent publications have provided evidence that ILC2s and ILC3s are indispensable for proper immune responses using different genetic models[13,16,30,31]. In particular, in the context of worm infections, ILC2s and Th2 cells fulfill complementary functions to protect against *Nippostronglylus brasiliensis* and *Trichuris muris* infections. Immunity to worm infections is mediated by the cytokine IL-13, which can be secreted by ILC2 or Th2 cells and induces a 'weep and sweep' reaction to expel the parasites. During *Trichuris muris* infections, ILC2s are an essential source of Areg for the expansion of Th2 cells to fight the infection[16,28,32,33]. For an in-depth discussion of ILC2 and Th2 responses during worm infection, the reader is kindly referred to a recent review article on this topic[34].

In addition to their pivotal role in immunity to worm infections, ILC2s are required for the development of eosinophils and lung eosinophilia during allergic lung inflammation[12,13,35]. Although not formally demonstrated, it is likely that eosinophils depend on ILC2-derived IL-5 because they are the main producers of IL-5 in the tissue[12] and eosinophils are strictly IL-5 dependent[14]. However, to which degree GM-CSF that is also produced by ILC2s could contribute to the eosinophil phenotype in ILC2-deficient mice remains to be investigated[3,12,13].

This study describes a second immune cell subset, B1 cells, which strictly requires ILC2s for proper development. Previous studies have shown that IL-5 and ILC2s could stimulate B1 cells in vitro or following adoptive transfer into alymphoid mice[4–6]. However, the importance of

ILC2s for B1 cells in lymphoreplete mice have remained under debate, given a functional overlap with T cells in producing type 2 effector cytokines. Using conditional knockout mice in ILC2s, our study causally demonstrates that ILC2s are strictly required as a source of IL-5 in lymphoreplete mice, which cannot be compensated by other cell types, including CD4^+^ T cells. To draw definite conclusions about the non-redundant functions of ILC2s, we performed several experiments to control for unintended targeting of other immune cell populations, which could affect B1 cells. For example, some expression of *Nmur1* was reported in approximately 25% of small intestinal eosinophils but not in eosinophils from other tissues examined using an independently generated *Nmur1*^Tdtomato-Cre^ line[36]. We did not detect *Nmur1*^iCre-eGFP^ activity in small intestinal eosinophils (Fig. S2D), demonstrating that eosinophils are not targeted in our mouse model. The discrepancy between the mouse lines might be explained by the relatively low expression levels of *Nmur1* in eosinophils and the differences in how the *Nmur1*^Cre^ mouse models were generated with the resulting expression levels of reporter genes. However, since eosinophils were reduced in ILC2-deficient mice due to the pivotal IL-5 provided by ILC2s[12,13], we aimed to exclude potential effects of eosinophils on B1 cells by using eosinophil-deficient Δ*dblGata* mice. Since Δ*dblGata* mice did not show differences in B1 cells (Fig. S2G–K), these data argue against an important contribution of eosinophils to the B1 cell phenotype in ILC2-deficient mice.

Since *Nmur1*^iCre-eGFP^ was reported in a subpopulation of tissue-resident Th2 cells after chronic worm infection[37], it is crucial to determine potential effects of T cells in our model. To this end, we detected abolished expression of IL-5 in ILC2s but not in CD4^+^ T cells in *Nmur1*^iCre-eGFP^ *Il5*^fl/fl^ mice, confirming specific deletion of *Il5* in ILC2s (Figs. 3C, S4D). Furthermore, deletion of IL-5 in T cells using *Cd4*^Cre^ *Il5*^fl/fl^ mice did not affect B1 cells, arguing that T cell-derived IL-5 is

dispensable for B1 cell development (Fig. 3N, O). We also confirmed the absence of $Nmur1^{iCre-eGFP}$ and $Id2$ expression in B1 and B2 cells (Fig. S2A–D) to exclude direct effects of our targeting strategy on B cells.

Collectively, these data indicate that ILC2s are essential for the development of both eosinophils and B1 cells, and therefore fulfill non-redundant functions by supporting the development of these two cell types. Therefore, our study defines unique functions of ILC2s establishing them as absolute requirement for the development of eosinophils and B1 cells with important consequences for our understanding of the multilayered organization of the innate and adaptive immune system. While our study establishes a non-redundant role of ILC2-derived IL-5 for B1 cells in mice, further studies need to investigate the role of ILC2 in regulating B cells in humans.

Release of the alarmins IL-33 and secretion of IL-5 by ILC2s emerge as an innate pathway for T-cell-independent production of antibodies. It is well-established that T-cell dependent antibody production in follicles requires CD40L signaling provided by the T cell. Expression of CD40L in T cells is controlled by the TCR, co-stimulation, and cytokine production, where the latter two are dependent on engagement of pattern recognition receptors in antigen-presenting cells by their ligands. Interestingly, pathogen-associated molecular patterns are considered danger signals similar to IL-33. In addition, most toll-like receptors and the IL-33 receptor use the signaling adapter Myd88 for signal transduction. Therefore, these data support IL-33 as a danger-associated molecular pattern regulating an innate pathway that has evolved probably prior to CD40L co-stimulation to support innate production of antibodies via IL-5 by innate lymphocytes.

## Methods

### Mouse strains

C57BL/6 mice (*Mus musculus*) were purchased from Janvier. $Nmur1^{iCre-eGFP[13,16]}$, $Id2^{flox/flox[38]}$, $Gata3^{flox/flox[39]}$, $Il5^{cre/cre[12]}$, $Il33^{-/-[40]}$, $Il17rb^{-/-[28]}$, $Tslpr^{-/-[41]}$, $Il1rl1^{flox/flox[42]}$, $Areg^{flox/flox[43]}$, $Il4/13^{flox/flox[44]}$, $Il6^{flox/flox[45]}$, $Rosa26^{flSTOP-RFP/flSTOP-RFP[46]}$, $Rosa26^{flSTOP-YFP/flSTOP-YFP[47]}$, $Id2^{CreERt2[48]}$, $Cd4^{Cre[49]}$ and $Zcwpw1^{-/-[50]}$ on a C57BL/6 background were bred locally at Charité. $Il5^{flox/flox}$ mice were generated by GemPharmatech on a C57BL/6 background. In brief, exon 1 and 2 of the $Il5$ gene were flanked by two LoxP sites and validated by Sanger sequencing. Δ*dblGata* mice on a *4Get* background and *4Get* controls were from David Voehringer. Sex and age-matched male and female animals, usually aged 7–14 weeks, were used for experiments if not otherwise indicated. We did not use blinding or randomization to assign animals to experimental groups. The sample size of experimental groups was not determined by statistical methods. All animal experiments were approved and are in accordance with the local animal care committees (LAGeSo Berlin).

### Cell isolation

Peritoneal lavage and thoracal lavage cells were isolated by flushing the peritoneal / thoracal cavity with 10, and 3 ml PBS, respectively. Omentum and mesenteric fat were incubated in RPMI 1640 medium (Gibco) supplemented with 1% BSA (Sigma-Aldrich), collagenase II (1 mg/ml; Sigma-Aldrich) and DNaseI (100 µg/ml) for 20 min on a shaker at 37 °C. Afterwards, cells were dissociated using a pasteur pipette, and filtered through a 70 µm cell strainer, spun down and aspirated. Spleens were chopped and filtered. Small intestine was removed, cleaned from remaining fat tissue and washed in ice-cold PBS. Peyer's patches were eliminated, small intestine was opened longitudinally and washed in ice-cold PBS. Dissociation of epithelial cells was performed by incubation on a shaker at 37 °C in HBSS (Sigma-Aldrich) containing 10 mM Hepes (Gibco) and 5 mM EDTA (Roboklon) two times for 15 min. After each step, samples were vortexed and the epithelial fraction discarded. Afterwards, remaining tissue was chopped into small pieces and enzymatic digestion was performed using Dispase (0.5 U/ml; Corning), Collagenase D (0.5 mg/ml; Roche) and DNaseI (100 µg/ml; Sigma-Aldrich). Leukocytes were further enriched by Percoll gradient centrifugation (GE Healthcare). Lungs were chopped and incubated in the enzyme cocktail described above for 40 min on a shaker at 37 °C. The remaining tissues were mashed with a syringe plunger and single cell suspensions were filtered through a 70 µm cell strainer. Leukocytes were then further enriched by Percoll gradient centrifugation.

### Flow cytometry and cell sorting

Dead cells were routinely excluded with Zombie Aqua Fixable Viability Dye (Biolegend) or SYTOX Blue Dead Cell Stain (Thermo Fisher Scientific). Single cell suspensions were incubated on ice with anti-CD16/CD32 antibody and the following conjugated antibodies in PBS ($Ca^{2+}$ and $Mg^{2+}$-free, Sigma-Aldrich): CD45 (30-F11 or 104), CD19 (6D5), CD45R (RA3-6B2), CD23 (B3B4), CD43 (S11), CD11b (M1/70), IgM (eB121-15F9 or X-54), IgD (11-26c (11-26)), CD5 (53-7.3), CD21/35 (7G6), KLRG1 (2F1 or MAFA), SCA1 (D7), ST2 (RMST2-33), CD90 (30H12), NK1.1 (PK136), NKp46 (29A1.4), c-kit (2B8), CCR6 (29-2L17), CD4 (GK1.5), CD8a (53-6.7), TCRγδ (GL3), CD127 (A7R34), CD64 (X54-5/7.1), CD11c (N418), Siglec-F (E50-2440). T cells were excluded by staining for TCRβ (H57-597). For staining of ILCs, lineage-positive cells were excluded by staining for CD3ε (145-2C11 or 500A2), CD5 (53-7.3), CD19 (1D3 or 6D5), FcεRI (Mar-1) and Ly6G (1A8). For intracellularly cytokine staining for IL-5 (TRFK5) and IL-13 (eBio13A), cells were fixed in Cytofix/Cytoperm (BD Bioscience) for 20 min on ice and the antibodies applied in PBS containing 0.5% Saponin (Sigma Aldrich). For staining Ki-67 (B56), Gata3 (L50-823) FoxP3 (FJK-16s) intracellularly the cells were fixed by using the Foxp3 transcription factor buffer set (Thermo Fisher Scientific).

Antibodies used in flow cytometry were purchased from Biolegend, Thermo Fisher Scientific, Miltenyi Biotec, or BD Bioscience. All flow cytometry experiments were acquired using a custom configuration Fortessa flow cytometer and the FACS Diva software (BD Biosciences) and were analyzed with FlowJo V9.9.3 or V10.6.2 software (TreeStar) or sort-purified by using a custom configuration FACSAria cell sorter (BD Biosciences).

### In vitro B1 cell culture

$10^4$ B1 cells (gated on Live $CD45^+$ $CD19^+$ $CD23^-$) from the peritoneal cavity were sort-purified from C57BL/6 wild type mice and cultured for 5 days in complete medium with the addition of B-cell activating factor (BAFF, R&D Systems). Recombinant IL-5 (10 ng ml$^{-1}$, R&D Systems), supernatant from stimulated (IL-33 and IL-7, 20 ng ml$^{-1}$ each) ILC2s precultured for 3 days, $10^4$ sort-purified LC2s from the small intestine (gated on Live $CD45^+$ $Lin^-$ $CD127^+$ $KLRG1^+$) were added. B1 cell numbers were analysed by flow cytometry 5 days after the culture.

### Cytokine measurement

Bulk purified immune cells were incubated in DMEM with high-glucose supplemented with 10% FCS, 10 mM Hepes, 1 mM sodium pyruvate, non-essential amino acids, 80 µM 2-mercaptoethanol, 2 mM glutamine, 100 U ml$^{-1}$ penicillin, 100 µg ml$^{-1}$ streptomycin (all from Gibco) and Brefeldin A (Sigma Aldrich) in 96-well U-bottom microtitre plates (Nunc) for 4 h at 37 °C and 5% $CO_2$. If indicated, the culture was supplemented with a cocktail of IL-2, IL-7, IL-25, IL-33 (Biolegend, 100 ng ml$^{-1}$, each) or phorbol 12-myristate 13-acetate (PMA, 1 µg ml$^{-1}$) and ionomycin (Invitrogen, 0.5 µg ml$^{-1}$).

### Quantitative real-time PCR

Sorted cells were homogenized in Trizol (Thermo Fisher Scientific) and stored at −80 °C. RNA was extracted with chloroform and RNA concentration was determined using a Nanodrop 2000 spectrophotometer (Thermo Fisher Scientific). Reverse transcription of total RNA was performed using the High Capacity cDNA Reverse Transcription kit according to the protocol provided by the manufacturer (Thermo Fisher Scientific). Reaction was detected on a QuantStudio 5

Real-Time PCR (Thermo Fisher Scientific) using SYBR Green Master Mix with *Ccnd2* (forward: 5′- CCTGGATGCTAGAGGTCTGTG-3′, reverse: 5′- GGCCTTAGTGTGATGGGGAA-3′), *Ass1* (forward: 5′-ACACCTCCTGCATCCTCGT-3′, reverse: 5′- GCTCACATCCTCAATGA ACACCT-3′), *Anax2* (forward: 5′- ATGTCTACTGTCCACGAAATCCT-3′, reverse: 5′- CGAAGTTGGTGTAGGGTTTGACT-3′), *Zcwpw1* (forward: 5′-GATGAAGAACCGGGCATTGTT -3′, reverse: 5′- GGCCTAGCTTA GA TGTCCCCA -3′). *Il5, Il4* and *Il13* expression was determined using Taqman Gene Expression Assay (Applied Biosystems) for *Il5* (Mm00439646_m1), for *Il4* (Mm00445259_m1) and *Il13* (Mm00434204_m1) expression. Gene expression was normalized to the housekeeping gene *Hprt1* (Mm00446968_m1) for Taqman and *Hprt1* (forward: 5′-GATACAGGCCAGACTTTGTTGG-3′, reverse: 5′-CAACAGGACTCCTCGTATTTGC-3′) for SYBR Green.

## Helminth infection
Third-stage larvae (L3) of *N. brasiliensis* were purified with a Baermann apparatus. After washing three times in PBS, larvae were counted and 500 purified larvae were injected subcutaneously in PBS. Mice were killed and blood serum samples were collected at 11 days post infection.

## Alternaria alternata treatment
For allergic asthma induction, 10 μg *A. Alternata* extract (Greer Laboratories) in PBS were administered intranasally on three consecutive days. Mice were killed seven days after initial administration, sera were collected and analyzed.

## Serum isotype concentrations
Isotype concentration in serum samples were determined by using the LEGENDplex mouse IgE assay and the LEGENDplex mouse immunoglobulin isotyping panel multiplex beads-based assay (Biolegend) according to the manufacture's protocol. Samples were recorded on a custom configuration Fortessa flow cytometer and the FACS Diva software (BD Biosciences) and the flow cytometry data files were analyzed using the Legendplex cloud-based analysis software suite (Biolegend).

## Single cell RNA- sequencing of B cells
Peritoneal lavage cells of either 5 *Id2*flox/flox mice or 5 *Nmur1*iCre-eGFP *Id2*flox/flox mice were pooled and sort-purified as live CD45+ TCRβ- IgM+ IgD+ as indicated in Fig. S3a.

## Library preparation and sequencing
B cells from *Nmur1*iCre-eGFP *Id2*flox/flox mice and littermate controls were applied to the 10X Genomics workflow for cell capturing and scRNA gene expression (GEX) library preparation using the Chromium Single Cell 5′ v2 Library & Gel Bead Kit (10x Genomics) [38]. Final GEX libraries were obtained after fragmentation, adapter ligation, and final Index PCR using the Single Index Kit TT Set A. Qubit HS DNA assay kit (Life Technologies) was used for library quantification and fragment sizes were determined using the Fragment Analyzer with the HS NGS Fragment Kit (1–6000 bp) (Agilent). Sequencing was performed on a NextSeq2000 device (Illumina) using a NextSeq 1000/2000 P2 reagent (200 cycles) with the recommended sequencing conditions for 5′ GEX libraries (read1: 26nt, read2: 90nt, index1: 10nt, index2: 10).

## Single-cell transcriptome
Raw sequence reads were processed using cellranger (version 5.0.0), including the default detection of intact cells. Mkfastq and count were used in default parameter settings for demultiplexing and quantification of gene expression. Refdata-cellranger-mm10-1.2.0 was used as reference. The number of expected cells was set to 3000.

The cellranger output was analyzed in R (version 4.0.3) using the Seurat package (version 4.0.2). The single cell transcriptomic profiles

for 12,000 KO cells and 12,000 WT cells were integrated, normalized, variable genes were detected and a uniform manifold approximation and projection (UMAP) was performed in default parameter settings using FindIntegrationAnchors, IntegrateData, FindVariableGenes, RunPCA and RunUMAP with 30 principle components. Expression values are represented as ln (10,000 * UMIsGene)/UMIsTotal + 1). Similar clusters were identified using shared nearest neighbor (SNN) modularity optimization, SNN resolutions ranging from 0.1 to 1.0 in 0.1 increments were computed, or gating was performed manually using the Loupe Browser (10X Genomics). Subsequently, clusters were annotated by projection of indicated genes on the UMAP to assign different cell types. Signature genes were identified using FindAllMarkers in default parameter settings. Heatmaps are based on z-transformed expression values for genes with significant differences to means in different clusters as judged by a Bonferroni corrected *P*-value (Wilcoxon rank sum Test) below 0.01 and a minimal absolute fold-change to the mean of log2(1.3).

## Ig gene analysis
Full length Ig gene sequences obtained from cellranger (version 7.2.0) were analyzed using Change-O (version 1.3.0), filtered for productive Ig heavy and light chain, and Ig gene frequencies were calculated using the alakazam package (version 1.2.1)[51]. Prior to all comparisons cell numbers were equalized for each sample.

## Histology and immunofluorescence microscopy
For confocal imaging, the tissue was fixed in 2% PFA at 4 °C for 4 h and immersed in 30% sucrose at 4 °C overnight. Samples were frozen in tissue tek (Sakura) and sectioned at −23 °C at a thickness of 14 micrometers. Dried sections were rehydrated with PBS, permeabilized and blocked in 0.5% Saponin and 10% FCS for 15 min and incubated in primary antibodies for 1 h at RT. After washing with PBS, secondary antibodies were incubated for 30 min at RT. Sections were washed and counterstained with DAPI (Thermo Fisher Scientific), finally mounted in Permafluor (Thermo Fisher Scientific) and imaged within 12 h. The following antibodies were used: CD5 (53-7.3) coupled to APC-Cy7 or Alexa Fluor 750, B220 (RA63B2) coupled to Alexa Fluor 700, KLRG1 (2F1) coupled to Alexa Fluor 647, CD43 (S11) coupled to Phycoerythrin (1:300), RFP (purified polyclonal, Rockland, 600-401-379) (1:1000), GFP (Thermo Fisher Scientific, A-21311) coupled to Alexa Fluor 488, Hamster IgG (Thermo Fisher Scientific, A-21451) coupled to Alexa Fluor 647, Rabbit IgG (Thermo Fisher Scientific, R37118) coupled to Alexa Fluor 488. Primary antibodies were diluted 1:100 if not mentioned otherwise, secondary antibodies were diluted 1:400. Nuclei were counterstained with DAPI, and images were acquired and stitched on a Leica Stellaris 8 equipped with a white light laser and a 405 diode laser using an HC PL APO CS2 20x air objective, NA 0.75.

## Statistical analysis
Data is plotted showing the mean +/− standard deviation. *P* values of data sets were determined by unpaired two-tailed Student's, Mann Whitney test, ordinary one-way ANOVA with Tukey's multiple comparisons test, with 95% confidence interval. Before-mentioned statistical tests were performed with Graph Pad Prism V9 software (GraphPad Software, Inc.). (*$p < 0.05$; **$p < 0.01$; ***$p < 0.001$; ****$p < 0.0001$ and ns not significant).

## Reporting summary
Further information on research design is available in the Nature Portfolio Reporting Summary linked to this article.

## Data availability
The RNA-seq data are deposited in the Sequence Read Archive (SRA) repository database under the submission number SUB14679600. https://www.ncbi.nlm.nih.gov/bioproject/PRJNA1152249

Additional data generated in this study are provided in the Supplementary Information and Source Data files are provided with this paper.

## Code availability
Data was analyzed using the standard Seurat 4.0.1 or 4.0.2 pipeline, or with the stated variations.

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

## Acknowledgements

We thank the Benjamin Franklin Flow Cytometry Facility for cell sort-
ing, the European Mouse Mutant Archive (EMMA) together with J.
Hidalgo, UC Davis and MMRCC repository for providing frozen material
of *Il6*^flox/flox, *Il1rl1*^flox/flox and *Areg*^flox/flox mice, A. Nakae (*Il33*^−/−), A. McKenzie
(*Il17rb*^−/−), and A. Lasorella (*Id2*^flox/flox) for providing mice. This work was
supported by grants from the Else Kröner-Fresenius-Stiftung (Else
Kröner Graduate School „Re-Thinking Health", 2020_EKPK.22 to K.F.T.),
the European Research Council Starting Grant (ERCEA; 803087 to
C.S.N.K.), the German Research Foundation (DFG; Project-ID
506620580 and EXC2151 – 390873048 to T.R.; Project-ID 259373024 –
CRC/TRR 167, FOR2599 project 5, B05 – Project-ID 375876048 – CRC/
TRR 241, – KL 2963/5-2, SPP1937 – KL 2963/2-1 and KL 2963/3-1 to
C.S.N.K.; SPP1937 DU1295/4-1 to C.U.D; RTG2570 project A7 to D.V.;
Clinical Research Unit KFO 5023 'BecauseY'/Project number
504745852 to A.A.K.), the Swiss National Science foundation (Grant-
ID: 184425 to M.O.J.), the US National Institutes of Health (DK126871,
AI151599, AI095466, AI095608, AI142213, AR070116, AI172027,
DK132244 to D.A.), LEO foundation, Cure for IBD, Jill Roberts Institute,
the Sanders Family, the Rosanne H. Silbermann Foundation, the Glenn
Greenberg and Linda Vester Foundation, the Crohn's and Colitis
Foundation and the Allen Discovery Center Program, a Paul G. Allen
Frontiers Group advised program of the Paul G. Allen Family Founda-
tion (all to D.A.) and the Federal Ministry of Education and Research
(BMBF, CONAN, TreAT, 01GL2401C to M.F.M.), the Leibniz Association
through the Leibniz Collaborative Excellence "TargArt" project to
M.F.M. and "ImpACt" project to M.F.M. and A.A.K. and by the Instru-
ment Grants INST 335/845-1 FUGG, INST 335/597-1 FUGG and INST
335/777-1 FUGG.

## Author contributions

K.F.T. and M.O.J. carried out most experiments and analyzed the data.
K.J.J., G.M.G., C.T., M.W., A.P., N.S., S.H. helped performing the experi-
ments. P.M.F. performed RNA-seq analysis. J.S., T.R. performed Ig gene
analysis. P.D., D.V., D.A., C.U.D., A.A.K., M.F.M. provided crucial input and
tools for the study. M.O.J. composed the figures. M.O.J. and C.S.N.K.
conceived the project and wrote the manuscript with input from all co-
authors.

## Funding

## Competing interests

D.A. has contributed to scientific advisory boards at Pfizer, Takeda, FARE,
and the KRF. The other authors declare no competing interests.

## Additional information

**Supplementary information** The online version contains
supplementary material available at

Christoph S. N. Klose.

**Peer review information** *Nature Communications* thanks Marina Cella,
Hergen Spits and the other, anonymous, reviewers for their contribution
to the peer review of this work. A peer review file is available.

[1]Charité – Universitätsmedizin Berlin, Corporate member of Freie Universität Berlin and Humboldt-Universität zu Berlin, Department of Microbiology,
Infectious Diseases and Immunology, Hindenburgdamm 30, Berlin, Germany. [2]Institute of Molecular Medicine and Experimental Immunology, University
Hospital Bonn, Bonn, Germany. [3]Deutsches Rheuma-Forschungszentrum (DRFZ), an Institute of the Leibniz Association, Berlin, Germany. [4]Department of
Infection Biology, University Hospital Erlangen and FAU Profile Center Immunomedicine (FAU I-MED), Friedrich-Alexander University Erlangen-Nuremberg,
Erlangen, Germany. [5]The Picower Institute for Learning and Memory, Department of Brain and Cognitive Sciences, Massachusetts Institute of Technology,
Cambridge, MA, USA. [6]Jill Roberts Institute for Research in Inflammatory Bowel Disease, Weill Cornell Medicine, Cornell University, New York, NY, USA.
[7]Friedman Center for Nutrition and Inflammation, Weill Cornell Medicine, Cornell University, New York, NY, USA. [8]Joan and Sanford I. Weill Department of
Medicine, Weill Cornell Medicine, Cornell University, New York, NY, USA. [9]Department of Microbiology and Immunology, Weill Cornell Medicine, Cornell
University, New York, NY, USA. [10]Allen Discovery Center for Neuroimmune Interactions, Weill Cornell Medicine, Cornell University, New York, NY, USA.
[11]German Center for Child and Adolescent Health (DZKJ), Partner Site Berlin, Berlin, Germany. [12]These authors contributed equally: Karoline F. Troch, Manuel
O. Jakob. ✉e-mail: christoph.klose@charite.de

