## [Transparent Peer Review file · Nature Communications]

Group 2 innate lymphoid cells are a non-redundant source of interleukin-5 required for development and function of murine B1 cells

Corresponding Author: Professor Christoph Klose

Version 0:

Reviewer comments:

Reviewer #1

(Remarks to the Author)

The paper by Troch et al reports on the role of ILC2 in proliferation and function of mouse B1 cells. It was found that this role is non-redundant and involves IL-33-induced production of IL-5. The data supporting a role of ILC2s in proliferation and survival of B1 cells are strong. However when looking at the function, ie antibody production per B1 cell there is no effect of ILC2 deletion. This is mentioned in the text (line 224-227) whereas the title and abstract seems to indicate that ILC2 are also required for the function of the B1 cells (line 43 and 44). This should be clarified. Apart from this the data support the conclusions of the authors.

The translational value of the data is limited. There is no evidence the IL-5 is important for human B1 development. Therefore the title should indicate that ILC2 are required for development of mouse B1 cells.

Although I realise that investigating this falls outside the scope of this paper it is recommended to include a few sentences discussing a possible role ILC2 in development of human B1 cells.

Reviewer #2

(Remarks to the Author)

In this elegant study Klose and colleagues demonstrate that B1 B cells at multiple body sites are dependent on IL-5 produced by ILC2s, thus confirming previous data that showed that IL-5-deficient and IL-5ra-deficient mice display dysregulated B1 B cell development.

The authors show by flow cytometry and scRNAseq that both B1a and B1b B cells are severely depleted in *Nmur1creId2fl/fl* mice, while B2 B cells are conserved. IL-5 from ILC2s is critical to promote the proliferation and survival of B1 B cells and the production of anti-phosphatidylcholine antibodies.

The data are clearly presented and support the conclusions.

In Fig. 2A and B it is evident that cluster 2 completely disappear in *Nmur1creId2fl/fl*, while cluster 3 is largely conserved. The authors should better present the scRNAseq data and show genes (in particular, immune genes) that discriminate between cluster 2 and 3. Specifically, has cluster 3 any selective expression of other cytokine receptors that could explain the partial IL-5 independence of cluster 3? These data could be critical to better understand the complex biology of B1 B cells.

Reviewer #3

(Remarks to the Author)

Summary: The manuscript describes the effects of ILC2s on B1 cell development and function. ILC2s stimulated through the IL33 receptor promotes B1 cell development and antibody production in response to IL5 production, which is not compensated by other immune cells. The effects of IL5 produced by ILC2 on B1 cells have already been described in several papers, as referenced by the authors (4-6). The manuscript is novel in the tools used, including *Nmur1cre* conditional mice, although this Reviewer has concerns about this model that need to be addressed. The manuscript reads

well, and the results are clear. Please see comments below.

Major points.

1) Authors combine flow cytometry and histology to analyze the interactions between ILC2s and B-cells. While it is clear that soluble factors can modulate B1 cell function, the data presented in Figure 3B suggests close proximity between B1 cells and ILC2s. Authors should investigate whether cell-cell interactions can contribute to B1 cell function and discuss the different hypothesis.

2) In Figure 1, authors should provide a full picture of the peritoneal lavage cellularity. Were B1 cells the only cells to be affected by the absence of ILC2s? Are other B cell populations that do not rely on IL-5 modulated? Would other populations (such as Tregs) that are potentially modulated interact with B1 cells?

3) *Nmur1* (PMID:16373672) and *id2* (PMID: 15701714) were both shown to be expressed by eosinophils. This highlights issues in specificity of the mouse model used. Authors need to discuss expression of *Nmur1* in other cells than ILC2s and if defects in these cells are observed in the mouse model used. In fact, human B-cells have *Nmur1* mRNA (PMID: 10999960). This is a major issue regarding specificity.

4) Along the same lines, *Nmur1* is upregulated in human lung T-cells, known contributors of IL-5 in the lungs. The *Nmur1* expression seems to be context and tissue dependent. Authors need to clarify that *Nmur1* expression is restricted to ILC2s in their experimental model and discuss these issues. Would there be a difference between *Nmur1* expression between murine and human models? What is the contribution of T-cell-mediated IL-5 in the model used?

5) Authors show that the effects of ILC2-driven IL-5 are IL-33-specific. It is not clear why, and these results are not discussed. Why would IL-25 and TSLP, who both induce IL-5 in ILC2s, not have similar effects on B1 cells?

6) "Due to the importance of IL-5_r signals and the particular reduction of IL5_r⁺ B1 cells in ILC2-deficient mice...". While this Reviewer understands the reason that IL5_r was chosen over other receptors was based on literature and data on IL5_r⁺ B-cells, ultimately a function for the receptors detailed in Figure 2K should not be excluded. For example, ILC2s produce IL-10, IL-6 and IL-4, whose receptors are listed to be expressed by B1 cells. How do the authors firmly exclude a contribution of these pathways? Also, Figure 2K is labeled as "cytokine receptors" but includes *Il16*, and all genes should be italicized in the Figure.

7) It is interesting that all IgG subclasses in *N. brasiliensis* challenged mice are markedly reduced compared to control mice. Were they performed simultaneously under the same conditions? Are there any possible explanation for this?

Minor comments.

Line 62. Define *Areg*

The notation and order of experimental organs in the figures are inconsistent and confusing, so make them easy to read.

Fig1G seems to be missing statistical tests.

In Figures 2J and 6E-F, representing *Ki-67* as mean fluorescence intensity (MFI) would enhance the clarity of the data.

There are some ambiguities in bar graphs' legends. In some bar graphs the name of the targeted cell is mentioned above the graphs, while in some others the name of the tissue is mentioned above the bar graphs. Also, some figures like Fig. 1G, it does not mention which cell is shown. This makes a hard time figure out which cell in which tissue the authors are trying to show in each figure. Authors can add the name of the cells to the Y axis of the graphs, as "%B1 cell in live CD45⁺" instead of just putting "% of live CD45⁺".

Fig 3A and 3E: It is better to write "% of live cells" and "% of live Lin⁻ cells" instead of "% of live" and "% of live Lin⁻".

Including fluorescence minus one (FMO) controls in flow cytometry graphs would provide essential gating controls for data interpretation.

Clarification regarding the color coding of expression levels in Figures 2C-E would aid in interpretation, particularly specifying the meaning of "1" and "-1" in the context of expression changes.

Reviewer #4

(Remarks to the Author)

Using elegant mouse genetic tools and single cell transcriptome sequencing, Torch et al. reported a non-redundant role of ILC2 for the development and function of B1 cells both at the steady state and during worms/fungi infections. Specifically, IL-33 signal is essential for IL-5 expression by ILC2s at the steady state, which is essential for the maintenance of B1 cells. Previously, ILC2s have been shown to promote B1 cell proliferation *in vitro* and *in vivo* under inflammatory condition. In addition, IL-33 has been believed to be dispensable for IL-5 production by ILC2 under homeostatic condition. Therefore, the findings from this research are surprising and interesting, which highlights that ILC2-derived IL-5 at the physiological level is

indispensable for sustaining B1 cell development. Several points need to be further addressed.

1. Figure 1g, please describe the exact time point of analysis in the figure legends. It is interesting to observe that the development of B1 was impaired in young mice before weaning (looks like 2 weeks from the time scale). ILC2s are supposed to be very few at the early age. Is the reduction of B1 cells in *Nmur1iCreId2f/f* mice at an early age due to ILC2-derived IL-5? Did the authors also observe a reduction of B1 cells in *Nmur1iCreId5f/f* mice and *Il5Cre/Cre* mice?
2. Figure 3: The authors found that B1 cells were decreased in *Il5Cre/Cre* compared with *Il5Cre/+* mice. Did the authors compare the level of B1 cells between *Il5Cre/+* and *Il5+/+* mice? As *Il5Cre/+* has been used as a tool to delete genes in ILC2s in the field, it could be informative to analyze if half-allele deficiency of *Il5* could cause a defect in B1 cells.
3. Figure 2g, Ki67 expression in B1 cells was not analyzed based on the gating *CD19+CD23-*. Therefore, it was not clear if the decreased of Ki67 was due to reduced proliferation capacity of B1 cell per se, or due to an overall reduction of B1 cells constituting the majority of proliferating cells in *CD19+* cells. Same questions remain for Figure 6E and 6F.
4. Figure 6: To corroborate that the deficiency of B1 cells in *Nmur1iCreId1r1f/f* is due to reduced IL-5, analysis of IL-5 expression should also be performed in ILC2s isolated from *Nmur1iCreId1r1f/f* and control mice.

Version 1:

Reviewer comments:

Reviewer #1

(Remarks to the Author)

The authors have adequately addressed my comments.

Reviewer #2

(Remarks to the Author)

The authors have fully addressed and resolved my concerns in their revision.

Reviewer #3

(Remarks to the Author)

Regarding NMUR1, the authors did not fully address my concerns. It is important to reiterate that several studies from different laboratories have employed sensitive techniques, suggesting that this receptor is expressed in various immune cells. Specifically, NMUR1 expression has been reported in eosinophils (37708282) and B-cells (10999960), as mentioned by the authors. Additionally, flow cytometry data (35810259, Figure 1B) indicate that NMUR1 is expressed in most immune cell types, except neutrophils. While this Reviewer appreciates the novel approach used to assess NMUR1 expression in B-cells and eosinophils, these findings appear to contradict the existing literature. How do the authors reconcile this discrepancy?

Furthermore, the approach used to investigate T-cells does not definitively exclude the possibility that NMUR1 is targeted by the NMUR1-Cre system in T-cells. Although the study excludes a role for T-cells in B1 cell function, it does not rule out NMUR1 expression in T-cells altogether (several studies report NMUR1 expression in T-cells: 38071753, 35810259).

To move forward, it is crucial to acknowledge the existing data in the literature and address any potential technical limitations or flaws in the mouse models or data generated by the group. If the authors choose this path, they must provide a clear and balanced discussion. This is essential for guiding future use of the NMUR1-Cre mice, as there appears to be some confusion and inconsistency in the field.

Reviewer #4

(Remarks to the Author)

Thank the authors for addressing my concerns. The manuscript has been significantly improved. Now the manuscript is suitable to be accepted for publication.

Version 2:

Reviewer comments:

Reviewer #3

(Remarks to the Author)

The authors have sufficiently addressed the concerns, and the manuscript is now more balanced, explicitly acknowledging the expression of NMUR1 on various immune cells. This crucial adjustment ensures that readers grasp the nonspecific nature of NMUR1 in murine models.

Response to the Editor's and Reviewers' Concerns

We are very grateful to the editor and reviewers for their constructive critique and for their enthusiasm concerning the significance of our work. The Reviewers raised relevant points of concern that we have addressed point-by-point below. We have performed rigorous revision of the manuscript, which we believe addressed all points that were raised. Our major claims remain unchanged, but the comments that have been raised have helped us clarify and substantiate our claims.

We hope the Editors and Reviewers will find this revised manuscript acceptable for publication.

REVIEWER COMMENTS

Reviewer #1 (Remarks to the Author):

The paper by Troch et al reports on the role of ILC2 in proliferation and function of mouse B1 cells. It was found that this role is non-redundant and involves IL-33-induced production of IL-5. The data supporting a role of ILC2s in proliferation and survival of B1 cells are strong. However when looking at the function, ie antibody production per B1 cell there is no effect of ILC2 deletion. This is mentioned in the text (line 224-227) whereas the title and abstract seems to indicate that ILC2 are also required for the function of the B1 cells (line 43 and 44). This should be clarified. Apart from this the data support the conclusions of the authors.

Response:

We thank you for the positive evaluation of our manuscript. We are happy to revise the manuscript based on your suggestions.

The translational value of the data is limited. There is no evidence the IL-5 is important for human B1 development. Therefore the title should indicate that ILC2 are required for development of mouse B1 cells.

Response:

We thank this reviewer for this comment. According to the reviewer's suggestion, we included "murine" in the title of the manuscript. However, the IgE data in worm and allergy models and phosphatidyl choline-specific rearrangements argue for functional consequences resulting from ILC2 - B1 cell interaction.

Although I realise that investigating this falls outside the scope of this paper it is recommended to include a few sentences discussing a possible role ILC2 in development of human B1 cells.

Response:

We have included a brief comment about ILC2 - B1 cell interaction in humans on page 15. Unfortunately, ILC2 - B interaction is much less investigated in humans, and we could not find enough literature for a proper discussion on that topic. I know Kazuyo Moro presented some data on that topic at the ILC meeting in Hawaii, but her study has not yet been published. Therefore, we would suggest limiting it to the comment we have included on page 15.

Reviewer #2 (Remarks to the Author):

In this elegant study Klose and colleagues demonstrate that B1 B cells at multiple body sites are dependent on IL-5 produced by ILC2s, thus confirming previous data that showed that IL-5-deficient and IL-5ra-deficient mice display dysregulated B1 B cell development.

The authors show by flow cytometry and scRNAseq that both B1a and B1b B cells are severely depleted in *Nmur1creId2fl/fl* mice, while B2 B cells are conserved. IL-5 from ILC2s is critical to promote the proliferation and survival of B1 B cells and the production of anti-phosphatidylcholine antibodies.

The data are clearly presented and support the conclusions.

Response:

We thank you for the positive evaluation of our manuscript. We are happy to improve the manuscript based on the reviewers' suggestions.

In Fig. 2A and B it is evident that cluster 2 completely disappear in *Nmur1creId2fl/fl*, while cluster 3 is largely conserved. The authors should better present the scRNAseq data and show genes (in particular, immune genes) that discriminate between cluster 2 and 3. Specifically, has cluster 3 any selective expression of other cytokine receptors that could explain the partial IL-5 independence of cluster 3? These data could be critical to better understand the complex biology of B1 B cells.

Response:

This is an excellent suggestion. To better understand the specific disappearance of cluster 2 in the absence of ILC2s and to identify functional differences between clusters 2 and 3, we screened for expressed cytokines/cytokine receptors in both clusters. However, we did not observe obvious differences, which could explain the dependence of other factors maintaining cluster 3 (Reviewer Figure 1A). Interestingly, and in line with our previous data, only cluster 0 and cluster 2 showed expression of the *Il5ra1*, both of which disappeared in the absence of the ILC2-derived *Il5* signals (Main Figure 2,3, Reviewer Figure 1B). These data further underpin

Reviewer Figure 1. B cells from the peritoneal cavity were FACS-sorted and single-cell sequenced including the B cell repertoire. **A**, Violin plot of expressed cytokine-receptors comparing clusters 2 and 3. **B**, Boxplot of the expression of the *Il5ra* in all B1 cell clusters. **C**, Boxplot of the expression of *Ighm* comparing clusters 2 and 3. **D,E**, Quantification of the most important phosphatidyl-specific heavy- and corresponding light chains, *Ighv11-2*, *Igkv14-126* and *Ighv12-3*, *Igkv4-91* comparing clusters 2 and 3.

the role of IL-5 in B1 cell maintenance of certain B1 cell subsets. We included these data in Figure S3F. While screening for functional differences, we observed that B1 cells in cluster 2 express higher amounts of *Ighm* and phosphatidylcholine-specific BCR rearrangements, suggesting that cluster 2 might fulfill different functions. We included these data in Figure S3

panels F,G, Figure S5 panels D,E and adjusted the text on page 8, lines 173-175, and on page 11, lines 243-244.

Reviewer #3 (Remarks to the Author):

Summary: The manuscript describes the effects of ILC2s on B1 cell development and function. ILC2s stimulated through the IL33 receptor promotes B1 cell development and antibody production in response to IL5 production, which is not compensated by other immune cells. The effects of IL5 produced by ILC2 on B1 cells have already been described in several papers, as referenced by the authors (4-6). The manuscript is novel in the tools used, including *Nmur1*cre conditional mice, although this Reviewer has concerns about this model that need to be addressed. The manuscript reads well, and the results are clear. Please see comments below.

Major points.

1) Authors combine flow cytometry and histology to analyze the interactions between ILC2s and B-cells. While it is clear that soluble factors can modulate B1 cell function, the data presented in Figure 3B suggests close proximity between B1 cells and ILC2s. Authors should investigate whether cell-cell interactions can contribute to B1 cell function and discuss the different hypothesis.

Response:

We thank you for this suggestion. To address this question, we sort-purified B1 cells and performed an *in vitro* culture under different conditions. First, we added medium supplemented with BAFF only to the cells, secondly recombinant Interleukin 5 (IL-5), thirdly supernatant from ILC2s precultured and stimulated for 3 days, and finally, we performed cocultures with sort-purified ILC2s. We observed an increased number of B1 cells in all conditions compared to medium control, suggesting that soluble factors induce B1 cell maintenance, and cell-cell contact is not strictly required. The close proximity of ILC2s and B1 cells may create a local milieu high of IL-5. We included these data in Figure S2L and adjusted the text on page 7, lines 136-140.

Reviewer Figure 2. A, 10^4 B1 cells (Live CD45⁺ CD19⁺ CD23⁻) from the peritoneal cavity were sort-purified from C57BL/6 wild type mice and cultured for 5 days in medium with the addition of B-cell activating factor (BAFF). Recombinant IL-5 (rIL-5), or supernatant (SN) from ILC2s precultured for 3 days *in vitro* or 10^4 ILC2s were added. B1 B cell numbers were analysed by flow cytometry.

2) In Figure 1, authors should provide a full picture of the peritoneal lavage cellularity. Were B1 cells the only cells to be affected by the absence of ILC2s? Are other B cell populations that do not rely on IL-5 modulated? Would other populations (such as Tregs) that are potentially modulated interact with B1 cells?

Response:

We performed detailed phenotyping of the peritoneal lavage in ILC2-deficient mice and compared them to co-housed littermate mice, as suggested by the reviewer. In the B2 cell compartment, we did not observe differences in relative and absolute cell numbers, suggesting that B2 cells are not dependent on ILC2 signals (Reviewer Figure 3A). Within the T cell compartment, we did not detect any differences in relative and absolute cell numbers, also when

gating CD4 or CD8 T cells separately (Reviewer Figure 3B). We also analyzed type 2 Tregs, which are producers of IL-5 in the T cell compartment. Also, no difference was detected here (Reviewer Figure 3C). When investigating ILCs, we did detect a slight relative increase in type 3 ILCs, which might be due to the missing competition of ILC2s. However, this relative increase did not translate in absolute cell numbers nor in differences in CCR6⁺ or NKp46⁺ ILC3 subsets (Reviewer Figure 3E-G). In sum, we did not detect any more differences in the peritoneal cavity in the absence of ILC2s. We included these data in Figure S1 panels H-P and adjusted the text on page 6, lines 117-119.

Reviewer Figure 3. Flow cytometric quantification of indicated cell subsets in the peritoneal lavage in *Nmur1^{iCre} Id2^{fl/fl}* (blue bars) and littermate *Id2^{fl/fl}* (white bars) mice. All cells were pre-gated on live CD45⁺. **A**, B2 cells were gated as TCRβ⁺ CD23⁺ CD19⁺. **B**, T cells were gated as CD3⁺ CD5⁺ +/- CD4⁺ / CD8⁺. **C**, Type 2 T regs were gated as CD3⁺ CD4⁺ Foxp3⁺ Gata3⁺ KLRG1⁺ ST2⁺. **D**, γδT cells were gated as CD3⁺ CD5⁺ TCRγδ⁺. **E-G**, ILCs cells were pre-gated on Lin⁻ (CD3 CD5 CD19 FCεR1a Ly6G) CD127⁺. ILC1 were further gated on NK1.1⁺ NKp46⁺. ILC3 were gated on NK1.1⁻ c-Kit⁺ and subclassified in NKp46⁺ or CCR6⁺. **H**, Macrophages were gated on Lin⁻ (CD3 CD5 CD19 FCεR1a NK1.1) CD11b⁺ CD64⁺. **I**, Neutrophils were gated on Lin⁻ (CD3 CD5 CD19 FCεR1a NK1.1) CD11b⁺ CD11c⁺ Ly6G⁺. **J**, Dendritic cells (DC) were gated on Lin⁻ (CD3 CD5 CD19 FCεR1a NK1.1) Ly6G⁻

3) *Nmur1* (PMID:16373672) and *id2* (PMID: 15701714) were both shown to be expressed by eosinophils. This highlights issues in specificity of the mouse model used. Authors need to discuss expression of *Nmur1* in other cells than ILC2s and if defects in these cells are observed in the mouse model used. In fact, human B-cells have *Nmur1* mRNA (PMID: 10999960). This is a major issue regarding specificity.

Response:

As suggested by this reviewer, we analyzed the *Nmur1* expression in eosinophils to exclude a potential effect of the used mouse model and the observed B1 cell phenotype. When using *Nmur1^{iCre} Rosa26^{flSTOP-tdRFP}* mice, we are able to fate-label all cells expressing *Nmur1*, such as ILC2 (Reviewer Figure 4A). However, we did not observe labelling of eosinophils (Reviewer Figure 4A), nor B1 cells or B2 cells (Reviewer Figure 4B), suggesting that these cells are not targeted. In addition, we did also not detect *Id2* expression in B1 cells (Reviewer Figure 4C). Although the fate-labeling studies clearly argue against any effects on targeting eosinophils or B1 cells, we aimed to formally exclude potential indirect effects of our targeting approach. Since ILC2s and IL-5 are strictly required for eosinophil development, we analysed *ΔdblGata* mice, which lack eosinophils but not ILC2s. Indeed, we did not find eosinophils in the peritoneal cavity in comparison to littermate control mice (Reviewer Figure 4D). To examine if eosinophils are required for B1 cell development, we analysed the peritoneal cavity and the

thoracic cavity by flow cytometry. We did not observe a reduction in B1 cells, providing direct evidence that eosinophils are not required for B1 cell development (Reviewer Figure 4E,F). In total, the data in the manuscript argue for a direct effect of ILC2-derived IL-5 on B1 cells without involvement of eosinophils or unintentional targeting of B cells using *Nmur1^{iCre} Id2^{fl/fl}* mice. We included all these data in Figure S2 and adjusted the text on page 6/7, lines 122-129.

Reviewer Figure 4. A,B, FACS-based quantification of the *Nmur1* fate-labelling signal from *Nmur1^{iCre}Rosa26^{flSTOP-tdRFP/+}* mice of indicated cell subsets. C, YFP expression in *Id2^{creErt2}Rosa26^{flSTOP-tdYFP/+}* mice after 5 weeks of Tamoxifen food administration of indicated cell subsets. D-F, Control (*4Get*) and *dblGata* (*4Get*) mice were analysed in steady state. D, Flow cytometric plots and quantification of Eosinophils pre-gated on live CD45⁺ in the peritoneal cavity. E, F, Flow cytometric plots and quantification of B1 cells in the E. peritoneal cavity and the F, thoracic lavage.

4) Along the same lines, *Nmur1* is upregulated in human lung T-cells, known contributors of IL-5 in the lungs. The *Nmur1* expression seems to be context and tissue dependent. Authors need to clarify that *Nmur1* expression is restricted to ILC2s in their experimental model and discuss these issues. Would there be a difference between *Nmur1* expression between murine and human models? What is the contribution of T-cell-mediated IL-5 in the model used?

Response:

We agree with the reviewer that it is highly relevant to exclude a potential contribution of T cell-derived IL-5 for B1 cells. To directly tackle this question genetically, we crossed *Il5^{fl/fl}* mice to *Cd4^{cre}* mice, to conditionally delete IL-5 in all T cells, but not in ILC2s. We analysed B1 cells in the peritoneal cavity and the thoracic lavage of *Cd4^{cre} Il5^{fl/fl}* mice. However, we did not observe a reduction of B1 cells in *Cd4^{cre} Il5^{fl/fl}* mice compared to littermate controls (Reviewer Figure 5A,B). Therefore, the data obtained from this newly generated mouse line demonstrate that T cell derived IL-5 is dispensable for development of B1 and support a non-redundant function of ILC2s in this process. These newly generated data indeed strengthen the manuscript. We included these data in Figure 3 panels N,O and adjusted the text on page 10, lines 212-217.

Reviewer Figure 5. Flow cytometric plots and quantification of B1 cells in *Cd4^{Cre} Il5^{fl/fl}* (purple) and littermate *Il5^{fl/fl}* mice (white) in the **A**, peritoneal cavity and **B**, and the thoracic lavage in steady state. B1 cells were pre-gated on live CD45⁺ TCRβ⁻. Each symbol represents

5) Authors show that the effects of ILC2-driven IL-5 are IL-33-specific. It is not clear why, and these results are not discussed. Why would IL-25 and TSLP, who both induce IL-5 in ILC2s, not have similar effects on B1 cells?

Response:

We thank this reviewer for bringing this to our attention. To validate our findings, we performed sort-purification of ILC2s from *Tslpr^{-/-}*, *Il17rb^{-/-}*, *Il33^{-/-}* and *Nmur1^{iCre+}St2^{fl/fl}* and compared the *Il5* expression to control mice. In line with our previous results, only the lack of IL-33 or the receptor ST2 reduced *Il5* expression, whereas the other alarmins did not significantly affect its expression. Given the different effects IL-25 and IL-33 have on ILC2 fate decisions, it is less surprising that *Il5* expression is IL-33 dependent. IL-25 secreted by tuft cells in the intestine promotes the generation of a subset of ILC2s coined inflammatory ILC2s, shown to promote worm expulsion of the gut-dwelling nematode *Nippostrongylus brasiliensis* via the production of IL-13(1-3). IL-13 is the main cytokine responsible for activating the epithelium to induce the so-called “weep and sweep reaction” to expel the parasite (4, 5). In contrast, IL-5 and the target cells, eosinophils and B1 cells are less important for immunity against *Nippostrongylus brasiliensis* (6, 7).

We included all these data in Figure 6 panel K and adjusted the text on page 13, lines 286-289.

Reviewer Figure 6. A, Sort-purified small intestinal ILC2s gated as Live CD45⁺ Lin⁻ (CD3, CD5, CD19, Fcεr1, Ly6G) CD127⁺ KLRG1⁺ of *Tslpr^{-/-}*, *Il17rb^{-/-}*, *Il33^{-/-}* and *Nmur1^{iCre+}Il1rl1^{fl/fl}* and control mice and qPCR

6) “Due to the importance of IL-5α signals and the particular reduction of IL5ra⁺ B1 cells in ILC2-deficient mice...”. While this Reviewer understands the reason that IL5ra was chosen over other receptors was based on literature and data on IL5ra⁺ B-cells, ultimately a function for the receptors detailed in Figure 2K should not be excluded. For example, ILC2s produce IL-10, IL-6 and IL-4, whose receptors are listed to be expressed by B1 cells. How do the authors firmly exclude a contribution of these pathways? Also, Figure 2K is labeled as “cytokine receptors” but includes Il16, and all genes should be italicized in the Figure.

Reviewer Figure 7. Flow cytometry plots and quantification of total B1, B1a and B1b cells (based on the marker CD5) of peritoneal lavage (PL), thoracic lavage (TL) and omentum of **A**, *Nmur1^{iCre} Il4/Il13^{fl/fl}* and littermate *Il4/Il13^{fl/fl}* control mice, **B**, *Nmur1^{iCre} Il6^{fl/fl}* and littermate *Il6^{fl/fl}* mice. **C**, 10⁴ B1 B cells (Live CD45⁺ CD19⁺ CD23⁻) from the peritoneal cavity were sort-purified from C57BL/6 wild type mice and cultured for 5 days in medium with the addition of B-cell activating factor (BAFF). Recombinant IL-10 (rIL-10) was added. B1 cell numbers were analysed by flow cytometry.

Response:

As suggested by the reviewer, we investigated the different cytokines expressed by ILC2 and studied their effect on the B1 cell compartment. To this end, we have generated two mouse lines: by crossing *Il4/Il13^{fl/fl}* with *Nmur1^{iCre}* mice, we deleted the cytokines IL-4 and IL-13 on ILC2, and by crossing *Il-6^{fl/fl}* with *Nmur1^{iCre}* mice, we abrogated the IL-6 secretion by ILC2. However, we did not observe a reduction in B1 cells, suggesting that these cytokines do not play a role in B1 cell maintenance (Reviewer Figure 7A,B). Further, we tested recombinant IL-10 on sort-purified B1 cells *in vitro* but did not observe any effects on B1 cell numbers (Reviewer Figure 7C). Taken together, these data do not support the idea that the cytokines IL-4/13, IL-6 or IL-10 are required for B1 cell development. We included these data in Figure S4E-G with the exception of the IL-10 stimulation. We would respectfully propose not to include the IL-10 data because they are challenging to integrate into the flow of the manuscript in the revised manuscript, but we would be happy to do so at the request of the Reviewer or the Editor.

7) It is interesting that all IgG subclasses in *N. brasiliensis* challenged mice are markedly reduced compared to control mice. Were they performed simultaneously under the same conditions? Are there any possible explanation for this?

Response:

All Ig subclasses were simultaneously detected by a multiplex assay (Legendplex). The sharp increase in IgE could explain the reduction in IgG subclasses.

Minor comments.

Line 62. Define Areg

Response:

Areg is now defined.

The notation and order of experimental organs in the figures are inconsistent and confusing, so make them easy to read.

Response:

We re-arranged the figures according to the reviewers' suggestion. The sequence of how the data is shown is: peritoneal lavage, thoracic lavage, omentum, lung and eventually spleen.

Fig1G seems to be missing statistical tests.

Response:

We now added the statistical test for each time point.

In Figures 2J and 6E-F, representing Ki-67 as mean fluorescence intensity (MFI) would enhance the clarity of the data.

Response:

We analysed the MFI of Ki-67⁺ cells of B1 cells in Figure 2J and 6E-F (data not shown). However, we did not find a difference since the increase of Ki-67 reflects the percentage of dividing cells. A difference in the amount of Ki-67 protein is not expected. We would respectfully propose that these data are not included in the revised manuscript but would be happy to do so at the request of the Reviewer or the Editor.

There are some ambiguities in bar graphs' legends. In some bar graphs the name of the targeted cell is mentioned above the graphs, while in some others the name of the tissue is mentioned above the bar graphs. Also, some figures like Fig. 1G, it does not mention which cell is shown. This makes a hard time figure out which cell in which tissue the authors are trying to show in each figure. Authors can add the name of the cells to the Y axis of the graphs, as “%B1 cell in live CD45⁺” instead of just putting “% of live CD45⁺”.

Response:

We followed the reviewers' suggestion and now define the cell type in the Y axis.

Fig 3A and 3E: It is better to write “% of live cells” and “% of live Lin- cells” instead of “% of live” and “% of live Lin-”.

Response:

We thank you for this comment. We changed the labels accordingly.

Including fluorescence minus one (FMO) controls in flow cytometry graphs would provide essential gating controls for data interpretation.

Response:

As suggested by you we performed FMO controls of the most important markers as well as the appropriate isotype control for Ki-67 (Reviewer Figure 9). These data confirm the specificity of our flow cytometry stainings. We included these data in Figure 1 panel D, Figure S3 panel E.

Clarification regarding the color coding of expression levels in Figures 2C-E would aid in interpretation, particularly specifying the meaning of "1" and "-1" in the context of expression changes.

Response:

All sequencing data were analyzed using the Seurat v4 package in R (<https://satijalab.org/seurat/>). The data were normalized using NormalizeData (logNormalize) and average expression was calculated using AverageExpression. Dotplot and Heatmaps were generated from scaled data. Average expression is the average of the normalized expression values. Scaled expression values centers the expression to have zero mean and a standard deviation of one.

“LogNormalize”: Feature counts for each cell are divided by the total counts for that cell and multiplied by the scale.factor. This is then natural-log transformed using log1p

AverageExpression : Returns a matrix with genes as rows, identity classes as columns.

ScaleData: Scales and centers features in the dataset. If variables are provided in vars.to.regress, they are individually regressed against each feature, and the resulting residuals are then scaled and centered.

Reviewer #4 (Remarks to the Author):

Using elegant mouse genetic tools and single cell transcriptome sequencing, Torch et al. reported a non-redundant role of ILC2 for the development and function of B1 cells both at the steady state and during worms/fungi infections. Specifically, IL-33 signal is essential for IL-5 expression by ILC2s at the steady state, which is essential for the maintenance of B1 cells. Previously, ILC2s have been shown to promote B1 cell proliferation in vitro and in vivo under inflammatory condition. In addition, IL-33 has been believed to be dispensable for IL-5 production by ILC2 under homeostatic condition. Therefore, the findings from this research are surprising and interesting, which highlights that ILC2-derived IL-5 at the physiological level is indispensable for sustaining B1 cell development. Several points need to be further addressed.

1. Figure 1g, please describe the exact time point of analysis in the figure legends. It is interesting to observe that the development of B1 was impaired in young mice before weaning (looks like 2 weeks from the time scale). ILC2s are supposed to be very few at the early age. Is the reduction of B1 cells in *Nmur1^{iCre}Id2^{f/f}* mice at an early age due to ILC2-derived IL-5? Did the authors also observe a reduction of B1 cells in *Nmur1^{iCre}Il5^{f/f}* mice and *Il5^{Cre}/Cre* mice?

Reviewer Figure 10. **A**, Flow cytometric plot of small intestinal ILC2s 2 weeks after birth. Cells were pre-gated on Live CD45⁺ Lin⁻ (CD3, CD5, CD19, Fcεr1, Ly6G). **B**, Flow cytometric plots and quantification of B1 cells in 2 weeks old *Nmur1^{iCre} Il5^{fl/fl}* mice and littermate controls.

Response:

We thank you for the opportunity to address this important issue. When studying ILC2 in two-week-old mice, ILC2 are already present in large numbers and express the classical markers KLRG1 and ST2 (Reviewer Figure 10A). To address if ILC2-derived IL-5 plays a role early in the establishment of the B1 cell pool, we analyzed B1 cells from mice at two weeks after birth from *Nmur1^{iCre} Il5^{fl/fl}* mice, as suggested by the reviewer. Already in two-weeks-old mice, we observed that the B1 cell pool is greatly reduced in relative and absolute cell numbers, arguing that ILC2 not only maintain, but also are important in the development of B1 cells (Reviewer Figure 10B). We included these data in Figure 3, panel M and adjusted the text on page 10, lines 208-212.

2. Figure 3: The authors found that B1 cells were decreased in *Il5^{Cre}/Cre* compared with *Il5^{Cre}/+* mice. Did the authors compare the level of B1 cells between *Il5^{Cre}/+* and *Il5^{+/+}* mice? As *Il5^{Cre}/+* has been used as a tool to delete genes in ILC2s in the field, it could be informative to analyze if half-allele deficiency of *Il5* could cause a defect in B1 cells.

Response:

We thank the reviewer for bringing this to our attention. We reformed side by side analysis of B1 cells in *I15^{+/+}*, *I15^{cre/+}* and *I15^{Cre/Cre}* mice. However, we did not consistently detect an intermediate phenotype in B1 cell counts in heterozygous *I15* mice, suggesting that one allele and the respective IL-5-production is enough for B1 cell maintenance (Reviewer Figure 11A,B). We included these data in Figure S4, panel A,B.

Reviewer Figure 11. A,B, Flow cytometric quantification of B1 cells in *I15^{+/+}*, *I15^{Cre/+}* (light yellow), and *I15^{Cre/Cre}* (yellow) from the peritoneal cavity. **A,** All B1 cells in relative and absolute numbers, **B,** CD11b⁺ B1 cells in relative and absolute numbers.

3. Figure 2g, Ki67 expression in B1 cells was not analyzed based on the gating CD19⁺CD23⁻. Therefore, it was not clear if the decreased of Ki67 was due to reduced proliferation capacity of B1 cell per se, or due to an overall reduction of B1 cells constituting the majority of proliferating cells in CD19⁺ cells. Same questions remain for Figure 6E and 6F.

Response:

We excuse us for not making this clear in the beginning. The cells were pre-gated on CD19⁺CD23⁻ cells and the gate shown in each of the mentioned figures is based on the just mentioned pre-gate. We clarified the gating strategy in the Figure legends.

4. Figure 6: To corroborate that the deficiency of B1 cells in *Nmur1^{iCre}Il1r1^{fl/fl}* is due to reduced IL-5, analysis of IL-5 expression should also be performed in ILC2s isolated from *Nmur1^{iCre}Il1r1^{fl/fl}* and control mice.

Response:

We thank the reviewer for this excellent suggestion. We performed qPCR of sort-purified ILC2s from *Nmur1^{iCre} Il1r1^{fl/fl}* mice and observed a similar reduction in *I15* expression compared to control mice as seen in *Il33^{-/-}* ILC2s. We included these data into Figure 6, Panel K and adjusted the text on page 13, line 286-289.

Reviewer Figure 12. A, Sort-purified small intestinal ILC2s gated as Live CD45⁺ Lin⁻ (CD3, CD5, CD19, Fcεr1, Ly6G) CD127⁺ KLRG1⁺ of *Il33^{-/-}* and *Nmur1^{iCre+}Il1r1^{fl/fl}* and appropriate control mice and qPCR for *I15* was performed.

References

1. Huang Y, Guo L, Qiu J, Chen X, Hu-Li J, Siebenlist U, et al. IL-25-responsive, lineage-negative KLRG1(hi) cells are multipotential 'inflammatory' type 2 innate lymphoid cells. *Nat Immunol.* 2015;16(2):161-9.
2. Huang Y, Mao K, Chen X, Sun MA, Kawabe T, Li W, et al. S1P-dependent interorgan trafficking of group 2 innate lymphoid cells supports host defense. *Science.* 2018;359(6371):114-9.
3. von Moltke J, Ji M, Liang HE, and Locksley RM. Tuft-cell-derived IL-25 regulates an intestinal ILC2-epithelial response circuit. *Nature.* 2016;529(7585):221-5.
4. McKenzie GJ, Fallon PG, Emson CL, Grecnis RK, and McKenzie AN. Simultaneous disruption of interleukin (IL)-4 and IL-13 defines individual roles in T helper cell type 2-mediated responses. *J Exp Med.* 1999;189(10):1565-72.
5. Cliffe LJ, Humphreys NE, Lane TE, Potten CS, Booth C, and Grecnis RK. Accelerated intestinal epithelial cell turnover: a new mechanism of parasite expulsion. *Science.* 2005;308(5727):1463-5.
6. Zaiss DMW, Pearce EJ, Artis D, McKenzie ANJ, and Klose CSN. Cooperation of ILC2s and T(H)2 cells in the expulsion of intestinal helminth parasites. *Nat Rev Immunol.* 2023.
7. Martin RK, Damle SR, Valentine YA, Zellner MP, James BN, Lownik JC, et al. B1 Cell IgE Impedes Mast Cell-Mediated Enhancement of Parasite Expulsion through B2 IgE Blockade. *Cell Rep.* 2018;22(7):1824-34.

REVIEWER COMMENTS

Reviewer #3 (Remarks to the Author):

Regarding NMUR1, the authors did not fully address my concerns. It is important to reiterate that several studies from different laboratories have employed sensitive techniques, suggesting that this receptor is expressed in various immune cells. Specifically, NMUR1 expression has been reported in eosinophils (37708282) and B-cells (10999960), as mentioned by the authors. Additionally, flow cytometry data (35810259, Figure 1B) indicate that NMUR1 is expressed in most immune cell types, except neutrophils. While this Reviewer appreciates the novel approach used to assess NMUR1 expression in B-cells and eosinophils, these findings appear to contradict the existing literature. How do the authors reconcile this discrepancy?

Furthermore, the approach used to investigate T-cells does not definitively exclude the possibility that NMUR1 is targeted by the NMUR1-Cre system in T-cells. Although the study excludes a role for T-cells in B1 cell function, it does not rule out NMUR1 expression in T-cells altogether (several studies report NMUR1 expression in T-cells: 38071753, 35810259).

To move forward, it is crucial to acknowledge the existing data in the literature and address any potential technical limitations or flaws in the mouse models or data generated by the group. If the authors choose this path, they must provide a clear and balanced discussion. This is essential for guiding future use of the NMUR1-Cre mice, as there appears to be some confusion and inconsistency in the field.

We thank the reviewer for the opportunity to clarify this point. *Nmur1* was first identified as an ILC2-specific transcript among ILCs by Marco Colonna's group in 2015 (Robinette et al. Nature Immunology 2015, Supplementary Table 1, PMID: 25621825)¹. This dataset was integrated into the publicly available Immgen consortium database (www.immgen.org) and can be compared to the expression of other immune cells, including B cells, T cells, and eosinophils mentioned by the reviewer.

We have compared *Nmur1* expression in ILC2, T cell subsets, B cell subsets, eosinophils and additional immune cells using these data (Reviewer Figure 1). The Immgen RNA-seq data show that *Nmur1* is, to a large degree, enriched in ILC2 but not in other immune cells (Reviewer Figure 1). In addition, the Immgen data consortium provides microarray data, which show essentially the same (Reviewer Figure 2). I would like to stress that the RNA-Seq and microarray data are unbiased analyses of sort-purified immune cells and large collections of data, in which my lab was not involved in the acquisition of or the analysis of. Furthermore, the data is accessible to everybody in case you would like to convince yourself (www.immgen.org).

The Immgen data are in line with the analyses of reporter mice, NMUR1 antibody staining, and qPCR from sort-purified immune cells in mice from several groups^{2, 3, 4, 5, 6}.

The following studies (PMID: 35810259, 10999960) cited by the reviewer measure human *Nmur1* and are therefore irrelevant to our mouse model. The reviewer might want to suggest that *Nmur1* might be upregulated under pathologic conditions comparable to the human data. However, our study is carried out mainly at steady state.

The study by Zheng et al. (PMID: 38071753) was carried out in mice but detected only 0,79% of *Nmur1*⁺ cells among CD8⁺ T cells (Figure 5A) in a disease model and further comments that the expression is even lower in CD4⁺ T cells. While I admit that the interpretation of the authors is that *Nmur1* is expressed by T cells, I would rather conclude from these data that the expression is marginal and hope the reviewer can agree on that. This would also be consistent with data from the Kuchroo lab, which determined a 2500x difference in *Nmur1* expression comparing ILC2 versus T_H2 cells (PMID: 28902842, Extended Data Figure 5A)⁵. However, after chronic worm infection, a subpopulation of tissue-resident T cells in fat tissue was reported to express *Nmur1*, which we will acknowledge in our discussion (PMID: 36240286)⁷.

The study by Li et al. created an *Nmur1* reporter different from ours⁸. In agreement with our data, they find that eosinophils in all tissues examined do not express *Nmur1* except of 25% of eosinophils in the small intestine⁸. However, the expression seems low, as indicated by qPCR data (approximately 0,005 relative to *Gapdh*, Fig.1D)⁸. A second single-cell sequencing study did not detect the expression of *Nmur1* in eosinophils⁹. Taken together, these data suggest that *Nmur1* expression is rather low and that there might be a discrepancy between the mouse models. We do not detect *Nmur1*^{iCre} activity in eosinophils but have included this aspect in our discussion.

While in light of the extensive evidence of unbiased expression analysis from the Immgen consortium database, we would respectfully suggest that *Nmur1* is rather selectively expressed in mice and, therefore, a valuable mouse model for the scientific community. However, expression of *Nmur1* in a subset of eosinophils or upregulation in a subset of T cells after chronic worm infection should be considered. Therefore, we agree with the reviewer that a more detailed discussion regarding the *Nmur1* expression in other murine immune subsets, together with our strategies to mitigate the risks of unspecific targeting and certifying our conclusions would strengthen the manuscript. The detailed discussion can be found on page 14 and 15 of the revised manuscript.

Reviewer Figure 1: Normalized expression of *Nmur1* in sort-purified immune cells measured by RNA sequencing. Data were obtained from the Immgen data browser (www.immgen.org).

Gene: *Nmur1* (ProbeSet ID: 10356335)

RMA normalized values (CAUTION: V2 datasets in yellow -- see Protocols)

Reviewer Figure 2: Normalized values of *Nmur1* RNA in sort-purified immune cells determined by microarray. Data were obtained from the Immgen data browser (www.immgen.org).

1. Robinette, M.L. *et al.* Transcriptional programs define molecular characteristics of innate lymphoid cell classes and subsets. *Nat Immunol* **16**, 306-317 (2015).
2. Klose, C.S.N. *et al.* The neuropeptide neuromedin U stimulates innate lymphoid cells and type 2 inflammation. *Nature* **549**, 282-286 (2017).
3. Wallrapp, A. *et al.* The neuropeptide NMU amplifies ILC2-driven allergic lung inflammation. *Nature* **549**, 351-356 (2017).
4. Cardoso, V. *et al.* Neuronal regulation of type 2 innate lymphoid cells via neuromedin U. *Nature* **549**, 277-281 (2017).
5. Jarick, K.J. *et al.* Non-redundant functions of group 2 innate lymphoid cells. *Nature* **611**, 794-800 (2022).
6. Tsou, A.M. *et al.* Neuropeptide regulation of non-redundant ILC2 responses at barrier surfaces. *Nature* **611**, 787-793 (2022).
7. Kabat, A.M. *et al.* Resident T(H)2 cells orchestrate adipose tissue remodeling at a site adjacent to infection. *Sci Immunol* **7**, eadd3263 (2022).
8. Li, Y. *et al.* Neuromedin U programs eosinophils to promote mucosal immunity of the small intestine. *Science* **381**, 1189-1196 (2023).
9. Gurtner, A. *et al.* Active eosinophils regulate host defence and immune responses in colitis. *Nature* **615**, 151-157 (2023).